# A mitochondria-anchored isoform of the actin-nucleating spire protein regulates mitochondrial division

Uri Manor[1†], Sadie Bartholomew[2†], Gonen Golani[3], Eric Christenson[4], Michael Kozlov[3], Henry Higgs[5], James Spudich[2], Jennifer Lippincott-Schwartz[1*]

[1]Cell Biology and Metabolism Program, Eunice Kennedy Shriver National Institute of Child Health and Human Development, Bethesda, United States; [2]Department of Biochemistry, Stanford University School of Medicine, Stanford, United States; [3]Department of Physiology and Pharmacology, Tel Aviv University, Tel Aviv, Israel; [4]Unit on Structural and Chemical Biology of Membrane Proteins, Eunice Kennedy Shriver National Institute of Child Health and Human Development, Bethesda, United States; [5]Department of Biochemistry, Geisel School of Medicine, Hanover, United States

**Abstract** Mitochondrial division, essential for survival in mammals, is enhanced by an inter-organellar process involving ER tubules encircling and constricting mitochondria. The force for constriction is thought to involve actin polymerization by the ER-anchored isoform of the formin protein inverted formin 2 (INF2). Unknown is the mechanism triggering INF2-mediated actin polymerization at ER-mitochondria intersections. We show that a novel isoform of the formin-binding, actin-nucleating protein Spire, Spire1C, localizes to mitochondria and directly links mitochondria to the actin cytoskeleton and the ER. Spire1C binds INF2 and promotes actin assembly on mitochondrial surfaces. Disrupting either Spire1C actin- or formin-binding activities reduces mitochondrial constriction and division. We propose Spire1C cooperates with INF2 to regulate actin assembly at ER-mitochondrial contacts. Simulations support this model's feasibility and demonstrate polymerizing actin filaments can induce mitochondrial constriction. Thus, Spire1C is optimally positioned to serve as a molecular hub that links mitochondria to actin and the ER for regulation of mitochondrial division.

*For correspondence: lippincj@mail.nih.gov

†These authors contributed equally to this work

## Introduction

Mitochondrial division is a complex process that is essential for survival in mammals (*Nunnari and Suomalainen, 2012*; *Archer, 2014*) and is facilitated by the actin cytoskeleton (*De Vos et al., 2005*; *DuBoff et al., 2012*; *Korobova et al., 2013*, *2014*; *Hatch et al., 2014*; *Li et al., 2015*). Two distinct steps define mitochondrial division - an initial constriction of mitochondrial membranes, followed by final membrane scission (*Friedman et al., 2011*; *Korobova et al., 2013*; *Murley et al., 2013*; *Korobova et al., 2014*). Scission is mediated by the dynamin-related protein, Drp1, which self-assembles on the surface of the mitochondrial outer membrane into helices that drive final mitochondrial division (*Lackner and Nunnari, 2009*; *Archer, 2014*). The initial constriction step narrows the mitochondrial tube diameter, which is necessary for Drp1 helix assembly (*Labrousse et al., 1999*; *Yoon et al., 2001*; *Legesse-Miller et al., 2003*; *Ingerman et al., 2005*; *Friedman et al., 2011*; *Mears et al., 2011*; *Murley et al., 2013*). This step is independent of Drp1 and occurs at ER-mitochondria intersection zones where ER tubules associate with and wrap around the mitochondrial outer membrane along the plane of

**eLife digest** Mitochondria are structures within cells that provide the energy to power many biological processes that are essential for complex life. These structures are also highly dynamic and go through cycles of fission (in which a single mitochondrion splits in two) and fusion (in which two mitochondria merge into one). These processes both maintain the correct number of mitochondria in a cell and remove damaged ones, and defects in either can result in many diseases.

Previous research had shown that mitochondria are in close contact with another cellular structure called the endoplasmic reticulum. The points of contact mark the sites where mitochondria undergo fission, as small tubes of the endoplasmic reticulum wrap around, and then constrict, to split a mitochondrion.

Other recent work revealed that a protein called INF2 is anchored on the endoplasmic reticulum where it promotes mitochondrial constriction. This protein builds actin subunits into long filaments that provide the force for constriction. However, it was not clear how INF2 became active, and whether there are proteins on mitochondria that interact with INF2 or actin.

Manor, Bartholomew et al. have now used a combination of microscopy-based methods and biochemical analysis to discover that a mitochondrial protein called Spire1C performs all of these roles. Spire1C is found on the outer membrane of mitochondria; it interacts with INF2 to drive the formation of actin filaments that constrict mitochondria. These results suggest that Spire1C bridges the endoplasmic reticulum with the network of actin filaments. Further experiments then showed that increasing Spire1C levels in cells resulted in the mitochondria becoming fragmented due to increased constriction. On the other hand, depleting Spire1C had the opposite effect and caused mitochondria to become unusually elongated. Following on from this work, the next challenge is to see if Spire1C is used differently or similarly in the different processes that involve mitochondrial fission.

mitochondrial division (*Friedman et al., 2011*; *Korobova et al., 2013*; *Murley et al., 2013*; *Korobova et al., 2014*). Along these zones, actin filaments polymerize, providing the force needed for constriction that floppy ER tubules lack (*Korobova et al., 2013*, *2014*; *Hatch et al., 2014*).

Inverted formin 2 (INF2) is a formin family protein that promotes actin filament polymerization in a regulated fashion (*Korobova et al., 2013*, *2014*). An ER-anchored splice isoform of INF2 (usually referred to as INF2-CAAX) (*Chhabra et al., 2009*; *Korobova et al., 2013*) has been shown to facilitate mitochondrial constriction and division via its actin polymerization activity (*Korobova et al., 2013*). Given INF2, but not actin assembly, is localized throughout the ER, how INF2-mediated actin assembly is specifically triggered at ER-mitochondria intersections to ensure mitochondrial division remains an open central question.

Spire proteins are membrane-binding actin-nucleators that interact with and regulate formin proteins (*Bosch et al., 2007*; *Quinlan et al., 2007*; *Pechlivanis et al., 2009*; *Kerkhoff, 2011*; *Pfender et al., 2011*; *Schuh, 2011*; *Vizcarra et al., 2011*; *Zeth et al., 2011*; *Quinlan, 2013*; *Montaville et al., 2014*). Synergistic promotion of actin assembly by Spire and formin proteins has been implicated in driving a variety of processes ranging from vesicle trafficking to DNA repair in the nucleus (*Pfender et al., 2011*; *Schuh, 2011*; *Montaville et al., 2014*; *Belin et al., 2015*). In these systems, membrane-associated Spire proteins nucleate actin filaments, which are then further polymerized by formin proteins, ultimately leading to actin-dependent translocation. Given these characteristics of Spire proteins, we set out to investigate whether any Spire proteins could be involved in helping promote INF2- and actin-dependent constriction and division of mitochondria at ER-mitochondria association zones.

Vertebrates have two known Spire genes, Spire1 and Spire2. Each Spire protein contains highly conserved domains with specific capabilities, including: four actin-monomer binding WH2 domains necessary for nucleating actin filaments; an mFYVE domain that binds to intracellular membranes and facilitates oligomerization (*Kerkhoff, 2011*; *Dietrich et al., 2013*); and an N-terminal KIND domain that serves to bind to and regulate formin proteins (*Bosch et al., 2007*; *Quinlan et al., 2007*; *Pechlivanis et al., 2009*; *Kerkhoff, 2011*; *Pfender et al., 2011*; *Vizcarra et al., 2011*; *Zeth et al., 2011*; *Quinlan, 2013*; *Montaville et al., 2014*). Although no Spire proteins have been shown previously to localize to mitochondria or the ER (*Dietrich et al., 2013*), here we report a previously uncharacterized alternate

splice-isoform of Spire1 (named Spire1C) that localizes to mitochondria, promotes actin assembly on mitochondrial surfaces, and interacts with ER-anchored INF2 to regulate mitochondrial constriction and division. Our results support a model where Spire1C and INF2 coordinately drive actin- and ER-dependent mitochondrial division. They also reveal that Spire1C directly links mitochondria to both the actin cytoskeleton and the ER.

## Results

### Spire1C's previously uncharacterized alternate exon, ExonC, is highly conserved

We identified and characterized a novel alternate splice-isoform of Spire1 that contains KIND and WH2 domains common to all Spire proteins, as well as a previously uncharacterized unique 58 amino acid alternate exon sequence (ExonC) (*Figure 1A* and see 'Materials and methods' and *Figure 1—figure supplements 1–3* for details on Spire1C cloning, probe generation, and sequence information). Because of its alternative ExonC sequence, we named the Spire1 isoform Spire1C (*Figure 1A*). After determining that Spire1C mRNA was present in multiple mouse tissues (*Figure 1—figure supplement 3*), we examined its presence and conservation among species. Using the UCSC Genome Browser, we searched for additional DNA or mRNA sequences that contain Spire1C (*Kent et al., 2002*). When compared to mouse, we found striking identity in ExonC for rat, rabbit, human, dog, elephant, opossum, platypus, and chicken. It is not found in the annotated zebrafish sequence. Compared to our amplified mouse sequence of 58 amino acids, human ExonC differs by 2 residues (96% identical), platypus by 8 residues (86% identical), and chicken by 12 residues (79% identical). Sequence homology is strongest among mammals, but chicken maintains conservation that is unlikely to be solely due to chance. Conservation among species extends beyond the coding exon and into the upstream and downstream intronic regions of the gene, stretching ~150 bases in the 3′ direction (data not shown). These conserved extensions are likely involved in splicing regulation of the exon. The high level of conservation within and surrounding ExonC suggests that its role is indispensible for the health of the organism.

### Spire1C's ExonC is necessary and sufficient for localization to mitochondria

When cells were transfected with a myc-tagged Spire1C construct, the protein showed extensive co-distribution with the mitochondrial marker mitoRFP (*Figure 1B*, myc-Spire1C). A polyclonal antibody generated against a peptide containing the unique 58 amino acids in ExonC (*Figure 1B* and *Figure 1—figure supplements 1–3*) also showed extensive mitochondria-specific labeling within cells (*Figure 1B*, α–ExonC) (see *Figure 1—figure supplements 2, 3* and 'Materials and methods' for additional information on α-ExonC). Testing the role of ExonC in mitochondrial targeting of Spire1C, we found that a GFP-tagged ExonC fusion protein robustly localized to mitochondria in expressing cells (*Figure 1B*, GFP-ExonC). By contrast, a myc-tagged Spire1C construct lacking ExonC never showed specific mitochondrial localization (*Figure 1B*, myc-Spire1ΔC). These results suggest that Spire1C is an endogenously expressed mitochondria-associated protein that targets to mitochondria via its ExonC domain.

### Spire1C localizes to the mitochondrial outer membrane with its formin- and actin-binding domains facing the cytoplasm

To examine whether Spire1C distributes on the surface or interior of mitochondria we compared the distribution of GFP-ExonC (marking Spire1C) with mitoRFP (marking the mitochondrial matrix) using structured illumination microscopy (SIM), which gives a twofold resolution improvement over conventional confocal imaging (*Allen et al., 2014*). GFP-ExonC labeling on mitochondria surrounded that of mitoRFP labeling (*Figure 2A*), suggesting Spire1C localizes to the periphery of mitochondria, most likely on the mitochondrial outer membrane.

To confirm Spire1C's mitochondrial outer membrane localization, we employed a fluorescence protease protection (FPP) assay (*Lorenz et al., 2006*), which can determine a protein's membrane topology (*Figure 2B*). GFP was fused to the N-terminus of Spire1C (GFP-Spire1C), the N-terminus of ExonC (GFP-ExonC), or to the C-terminus of ExonC (ExonC-GFP) (*Figure 2C*). The constructs were

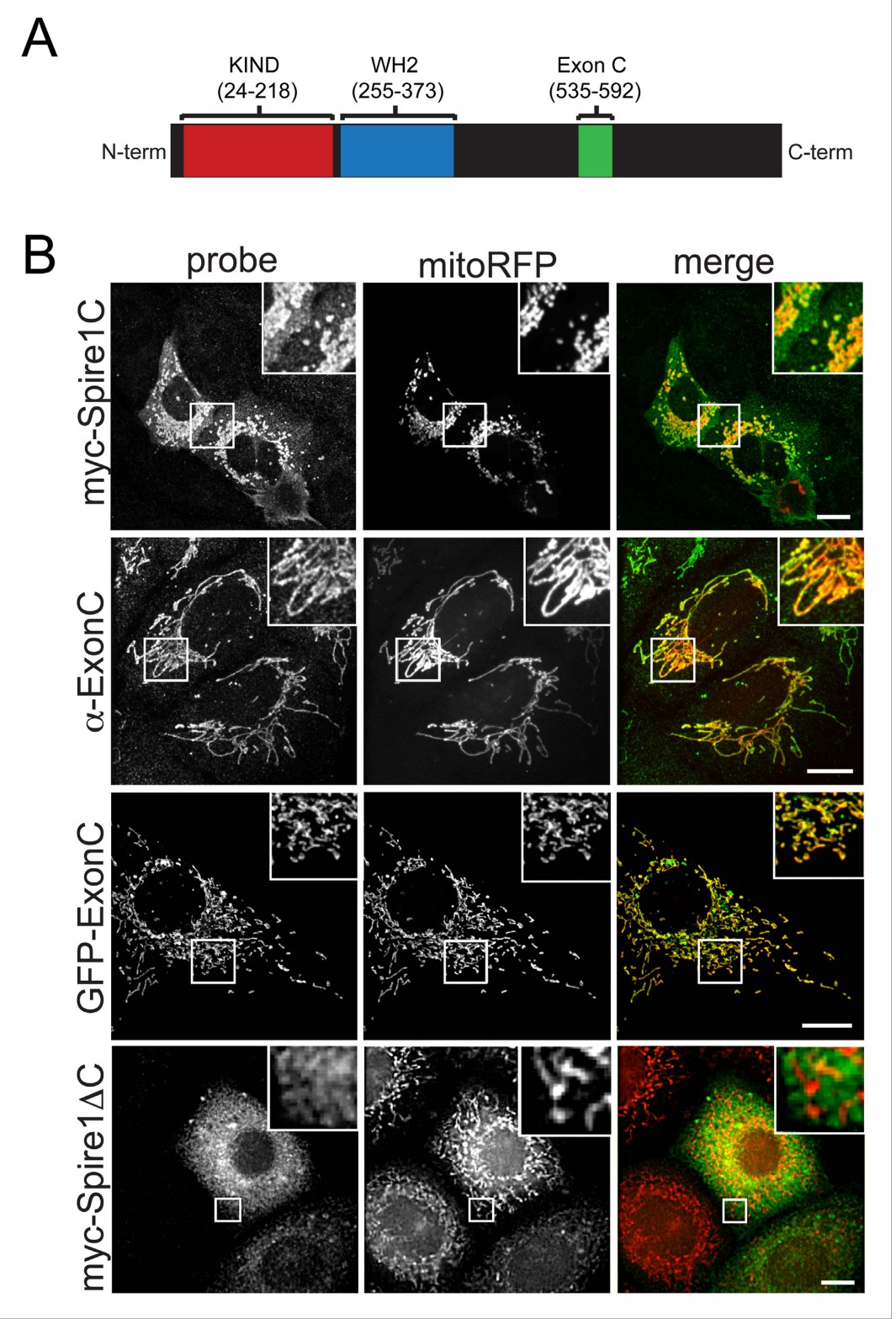

**Figure 1**. The Spire1 alternate exon ExonC is necessary and sufficient for localization to mitochondria. (**A**) Full length Spire1C domain structure: The number ranges indicate the amino acid regions of the conserved domains probed in this study. (**B**) Spire1C localizes to mitochondria. myc-Spire1C: U2OS cells cotransfected with myc-Spire1C and mitoRFP show robust localization of myc-Spire1C to mitochondria. α-ExonC: U2OS cells stained with an antibody raised against ExonC (α-ExonC) and expressing mitoRFP show endogenous Spire1C labeling on mitochondria.
*Figure 1. continued on next page*

*Figure 1. Continued*

GFP-ExonC: U2OS cells cotransfected with GFP-ExonC and mitoRFP show robust targeting of GFP-ExonC to mitochondria. myc-Spire1ΔC: U2OS cells cotransfected with myc-Spire1ΔC and mitoRFP show no specific targeting of myc-Spire1ΔC to mitochondria. All cells were fixed and primary antibodies were counterstained with Alexa-488 secondary antibody before imaging with confocal fluorescence microscopy. Scale bars: 10 µm. Inserts are magnifications of the boxed regions.

The following figure supplements are available for figure 1:

**Figure supplement 1**. Construction of the complete Spire1C protein sequence as explained in detail in the 'Materials and methods'.

**Figure supplement 2**. Constructs used to probe Spire1 function.

**Figure supplement 3**. Spire1C contains a previously uncharacterized alternate exon of 58 amino acids.

then co-expressed in cells with OMI-mCherry, a mitochondrial intermembrane space (IMS) protein (*Muñoz-Pinedo et al., 2006*). Thereafter, the plasma membrane of the cells was gently permeabilized with digitonin, followed by treatment with trypsin to extinguish cytoplasmic GFP fluorescence. Because mitochondrial membranes are not permeabilized by digitonin treatment, we reasoned that only if the fluorescent tag from these constructs faced the cytoplasm would their fluorescence be abolished by the trypsin. Both GFP-Spire1C and GFP-ExonC lost nearly all their fluorescence within 60 seconds of trypsin treatment. By contrast, fluorescence from ExonC-GFP was protected, similar to that seen for co-expressed OMI-mCherry, which as a mitochondrial IMS protein should be insensitive to trypsin (*Muñoz-Pinedo et al., 2006*) (*Figure 2D*; *Videos 1–3*). These results suggest that the WH2 and KIND domains of Spire1C face the cytoplasm (since both reside N-terminal to ExonC) (*Figure 2C*), a topology where they could participate in formin-binding and actin-nucleation activity. The data further suggest that the C-terminus of ExonC is not exposed to the cytoplasm. This raises the possibility that ExonC is embedded in the outer membrane, either as a transmembrane or hairpin protein.

In support of this notion, transmembrane domain prediction software (*Claros and von Heijne, 1994*) indicated that residues 26–46 within ExonC form an α-helix characteristic of prokaryotic transmembrane domains. Secondary structure prediction software PHYRE (*Kelley and Sternberg, 2009*) also predicted a second α-helix within ExonC (*Figure 1—figure supplement 3*). Interestingly, when we expressed a full-length Spire1C construct with GFP fused to the C-terminus (Spire1C-GFP), the protein remained mostly cytoplasmic and no longer properly localized to mitochondria (data not shown). The protein also rapidly escaped cells treated with digitonin, suggesting it was not able to target efficiently to mitochondria. Taken together, our data suggests that Spire1C is a mitochondrial outer membrane protein, possibly with a hairpin conformation given ExonC's two predicted α-helix domains (*Figure 1—figure supplement 3*).

We next performed photobleaching experiments to investigate the dynamics of Spire1C's association with mitochondrial membranes. Photobleaching of a portion of a mitochondrial element that expressed GFP-Spire1C resulted in rapid recovery of fluorescence, with replenishment arising first in regions close to the bleach site and later at regions further away (*Figure 2E* and *Figure 2—figure supplement 1*), similar to that seen for GFP-tagged proteins that feely diffuse along membranes (*Cole et al., 1996*). In contrast, very little recovery during the same time period occurred when an entire mitochondrial element expressing GFP-Spire1C was photobleached (*Figure 2F*). Thus, Spire1C appears to diffuse laterally along mitochondrial membranes, rather than rapidly bind and dissociate from the membrane, further supporting the idea of Spire1C being a mitochondrial outer membrane protein.

## Spire1C promotes actin assembly on mitochondrial surfaces

Given that Spire proteins can nucleate actin via their highly conserved WH2-repeat domain (*Quinlan et al., 2005*; *Loomis et al., 2006*; *Salles et al., 2009*), we next investigated whether overexpression of Spire1C promotes actin assembly on mitochondria. In non-transfected cells, only low levels of actin

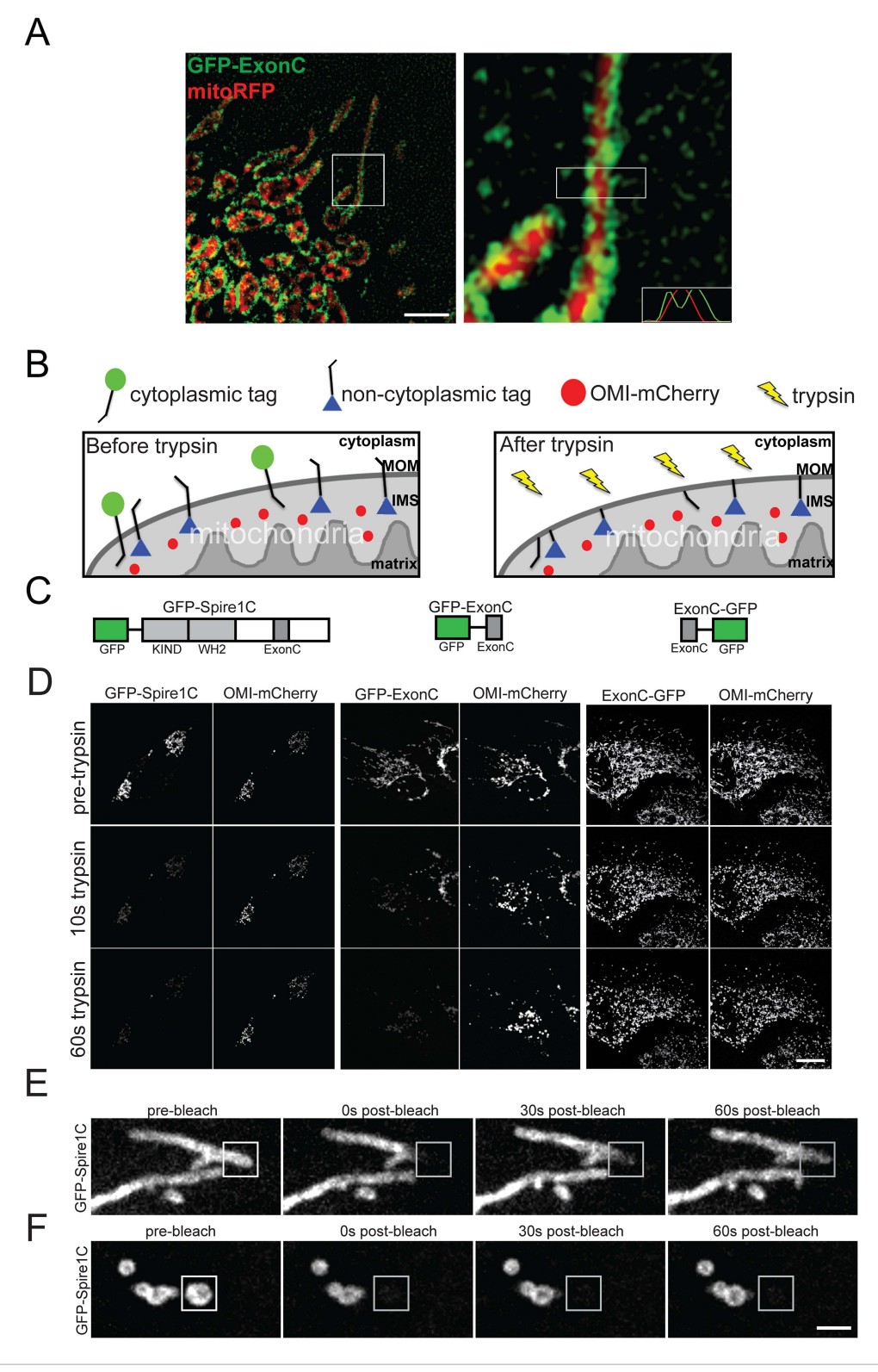

**Figure 2**. Spire1C localizes to the mitochondrial outer membrane with its formin and actin binding domains facing the cytoplasm. (**A**) ExonC localizes to the peripheral region of mitochondria. Left: structured illumination microscopy (SIM) image of a U2OS cell transfected with GFP-ExonC and mitoRFP reveals localization of GFP-ExonC to the periphery of mitochondria. Right: Magnification of boxed region on left. The insert on the lower right corner is
*Figure 2. continued on next page*

*Figure 2. Continued*

a fluorescence intensity linescan of the rectangular boxed region indicating the inversely related profiles of GFP-ExonC vs mitoRFP. Scale bar: 10 µm. (**B**) Illustration of the principle of the fluorescence protease protection (FPP) assay performed on mitochondrial outer membrane (MOM) proteins. If a fluorescent protein tag faces the cytoplasm (green circle), it is degraded by trypsin and its fluorescence is depleted. If the protein tag faces the interior of the mitochondria (blue triangle), it is protected from trypsin in the cytoplasm, and thus its fluorescence remains after trypsin addition. An intermembrane space (IMS) marker, OMI-mCherry, serves as a control to verify that the mitochondrial outer membrane has not been permeabilized by the digitonin treatment, and furthermore to confirm that trypsin is not degrading proteins in the IMS. (**C**) Schematic of the constructs used in our FPP assays. (**D**) Cells cotransfected with OMI-mCherry and GFP-Spire1C (N-terminal GFP tag, left) or GFP-ExonC (N-terminal GFP-tag, middle) were treated with 20 µM digitonin and 4 mM trypsin. In both cases OMI-mCherry fluorescence remained, whereas the N-terminal GFP tags were mostly depleted within 60 s after trypsin treatment. In contrast, in cells transfected with ExonC-GFP (C-terminus GFP tag), GFP fluorescence remains unchanged after 60 s of trypsin treatment. Scale bar: 10 µm. (**E**) GFP-Spire1C laterally diffuses along the mitochondrial outer membrane. A small region of a mitochondrion labeled with GFP-Spire1C was photobleached (white boxed region). The rapid, directional recovery from the unbleached region into the bleached region (see also Figure 2—figure supplement 2) suggests GFP-Spire1C is stably associated with and laterally diffuses along the mitochondrial outer membrane. (**F**) GFP-Spire1C does not readily exchange with the cytoplasm or neighboring mitochondria since photobleaching of an entire mitochondrion resulted in very low fluorescence recovery over the same period of time as in (**E**). Scale bar: 1 µm.

The following figure supplement is available for figure 2:

**Figure supplement 1**. GFP-Spire1C laterally diffuses on the mitochondrial outer membrane.

---

co-localized with mitochondria, without any apparent specificity (*Figure 3* and *Figure 3—figure supplement 1*, control). Upon Spire1C overexpression, however, actin accumulated to high levels specifically on mitochondria (*Figure 3* and *Figure 3—figure supplement 1*, Spire1C overexpression).

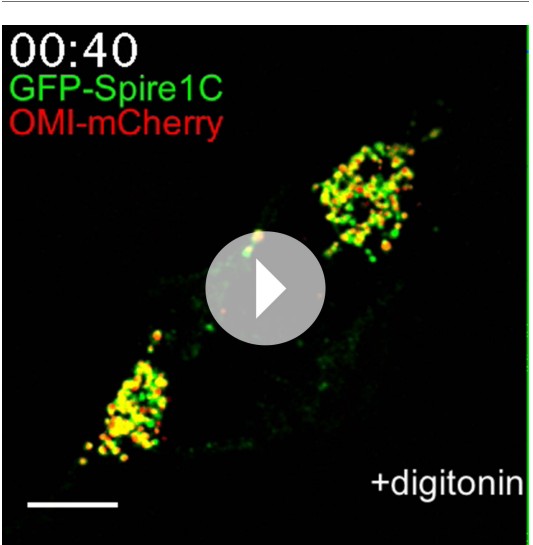

**Video 1.** A U2OS cell coexpressing GFP-Spire1C (N-terminus tag) and OMI-mCherry displays rapid loss of GFP fluorescence signal after the addition of 10 µM digitonin and 4 mM trypsin, whereas mCherry fluorescence persists, indicating that trypsin is degrading the GFP tag on the N-terminus of Spire1C in the cytoplasm, but not OMI-mCherry, which resides in the mitochondria intermembrane space (IMS). Scale bar: 10 µm.

This actin enrichment on mitochondrial surfaces was not dependent on Spire's formin-binding KIND domain, since overexpression of Spire1C lacking the KIND domain (Spire1CΔKIND) still induced actin enrichment on mitochondrial surfaces (*Figure 3*, Spire1CΔKIND overexpression). In contrast, actin enrichment on mitochondria was muted upon overexpression of Spire1C mWH2 (*Figure 3* and *Figure 3—figure supplement 1*, Spire1C mWH2 overexpression), which contains mutations in its WH2 domains that block Spire-mediated actin nucleation (*Loomis et al., 2006*; *Salles et al., 2009*) (see 'Materials and methods'). These results demonstrated that Spire1C promotes actin assembly on mitochondria, most likely through Spire1C's ability to nucleate actin filaments.

## Spire1C modulates mitochondrial fission via its formin-binding KIND and actin-nucleating WH2-repeat domains

Given that actin assembly has been shown to play an important role in regulating mitochondrial fission (*De Vos et al., 2005*; *DuBoff et al., 2012*; *Korobova et al., 2013*, *2014*; *Hatch et al., 2014*; *Li et al., 2015*), the observation that Spire1C

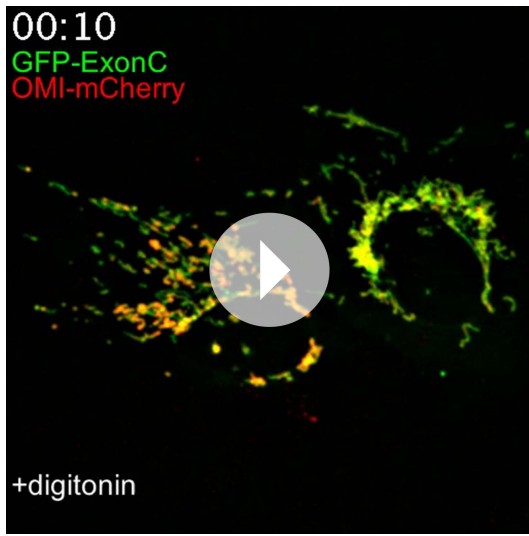

**Video 2.** A U2OS cell coexpressing GFP-ExonC (N-terminus tag) and OMI-mCherry displays rapid loss of GFP fluorescence signal after the addition of 10 μM digitonin and 4 mM trypsin, whereas mCherry fluorescence persists, indicating that trypsin is degrading the GFP tag on the N-terminus of Spire1C in the cytoplasm, but not OMI-mCherry, which resides in the mitochondria IMS. Scale bar: 10 μm.

**Video 3.** A U2OS cell coexpressing ExonC-GFP (C-terminus tag) and OMI-mCherry displays no loss of GFP fluorescence signal after the addition of 10 μM digitonin and 4 mM trypsin, indicating the GFP tag on the C-terminus of ExonC is protected within the mitochondrial lumen.
 Scale bar: 10 μm.

localizes to mitochondria and promotes actin assembly on mitochondrial surfaces raised the possibility that Spire1C plays a role in mitochondrial fission. To test this directly, we assessed the effect of Spire1C overexpression, mutation and depletion on mitochondrial morphology, length, and fission. While all cells in all conditions in this study displayed a combination of fragmented and tubular mitochondria, overexpression of Spire1C resulted in a significant shift towards fragmented mitochondria compared to control cells (*Figure 4A*, +Spire1C). In contrast, overexpression of Spire1C mWH2 or Spire1CΔKIND resulted in a shift towards tubular mitochondria (*Figure 4A*). Depletion of Spire1C using shRNA also resulted in a shift towards tubular mitochondria (*Figure 4B*). To quantify these changes, we measured mitochondrial lengths in each of these conditions, and found that Spire1C overexpression resulted in shorter mitochondria on average, whereas mutation or depletion of Spire1C resulted in longer mitochondria (*Figure 4C*, left graph, and *Figure 4—figure supplement 1*), resembling the effect of disrupting mitochondrial fission due to the depletion of Drp1 or INF2 from cells (*Bleazard et al., 1999*; *Labrousse et al., 1999*; *Wakabayashi et al., 2009*; *Korobova et al., 2013, 2014*; *Hatch et al., 2014*). To determine whether these morphological changes were a result of altered mitochondrial fission dynamics, we counted the number of mitochondrial fission events in each of these conditions. We found that fission events increased in frequency when cells were overexpressing Spire1C, whereas overexpression of Spire1C mWH2 or Spire1CΔKIND decreased the frequency of fission events (*Figure 4C*, right graph). Similarly, shRNA-mediated knockdown of Spire1C decreased the frequency of fission events (*Figure 4C*, right graph). Taken together, our results show Spire1C promotes mitochondrial fission via Spire1C's actin-nucleating and formin-binding capabilities.

## Disrupting the Spire1C KIND domain decreases ER-mitochondria overlaps

Since ER tubules have been implicated in mitochondrial constriction and division (*Friedman et al., 2011*; *Korobova et al., 2013*; *Murley et al., 2013*; *Korobova et al., 2014*), we investigated whether Spire1C influences ER-mitochondria association. Upon overexpression of Spire1C or Spire1C mWH2, we observed no significant change in ER-mitochondrial overlap or crossover sites of ER tubules and mitochondria compared to control cells (*Figure 5A,B*). However, in cells overexpressing Spire1CΔKIND,

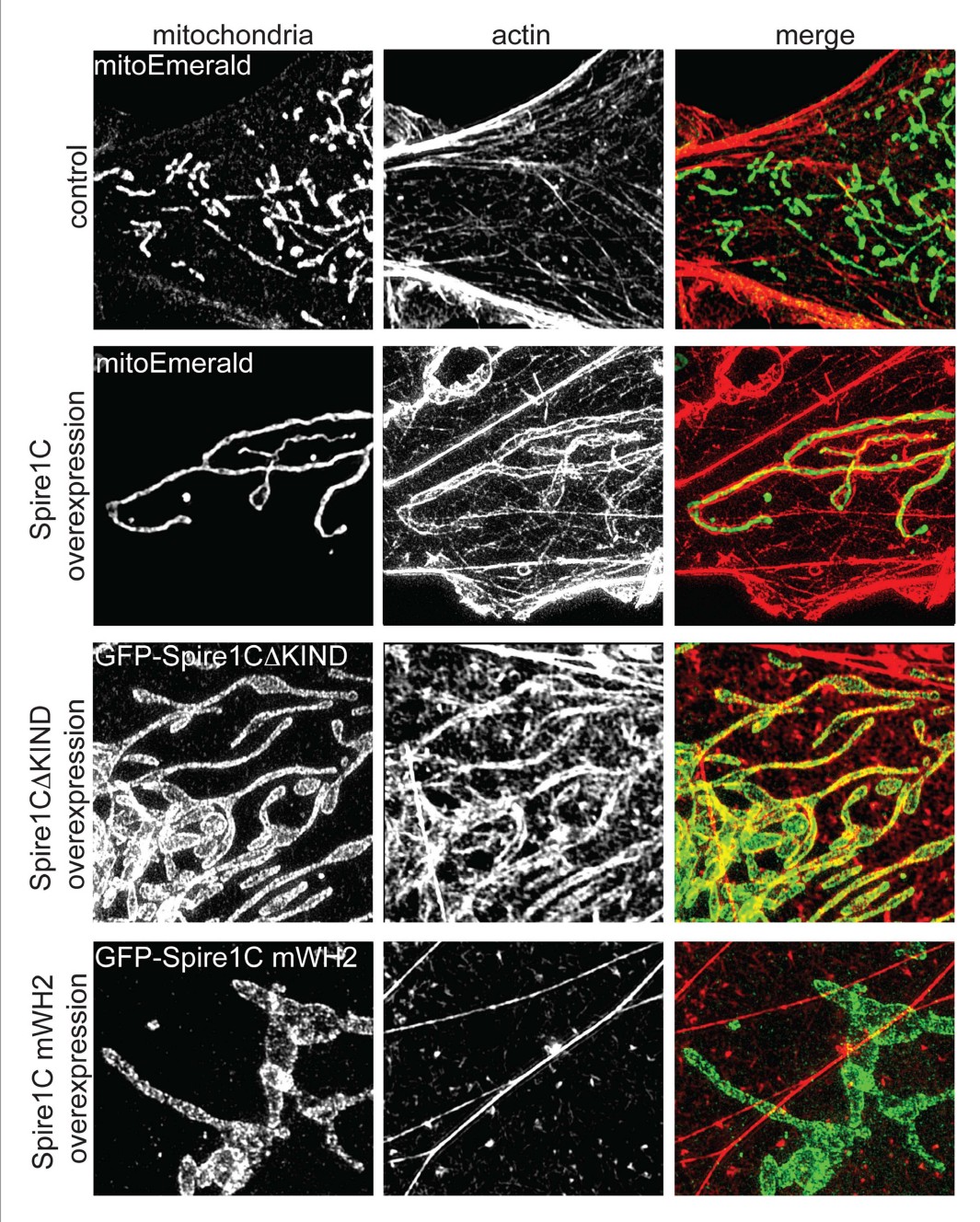

**Figure 3**. Spire1C promotes actin assembly on mitochondrial surfaces. Overexpression of Spire1C causes actin accumulation on mitochondria. Control: SIM image of a Cos7 cell expressing mitoEmerald and stained with phalloidin-568 to visualize actin shows low amounts of overlap between mitochondria and actin (Mander's: $0.43 \pm 0.020$, $n_{cells} = 26$). Spire1C overexpression: A Cos7 cell expressing mitoEmerald and overexpressing myc-Spire1C and stained with phalloidin-568 reveals significantly increased actin accumulation on mitochondria (Mander's: $0.64 \pm 0.066$, $n_{cells} = 19$, $p < 0.05$) compared to control cells. GFP-Spire1CΔKIND: A Cos7 cell overexpressing the formin-binding deficient GFP-Spire1CΔKIND stained with phalloidin-568 reveals significant accumulation of actin on mitochondria compared to control cells (Mander's: $0.55 \pm 0.017$, $n_{cells} = 18$, $p < 0.05$). A Cos7 cell overexpressing GFP-Spire1C mWH2 displays no increased accumulation of actin (Mander's: $0.41 \pm 0.040$, $n_{cells} = 15$, $p = 0.32$) compared to control cells. Scale bar: 5 μm.

The following figure supplement is available for figure 3:

**Figure supplement 1**. Spire1C promotes actin assembly near mitochondria in a WH2-dependent fashion.

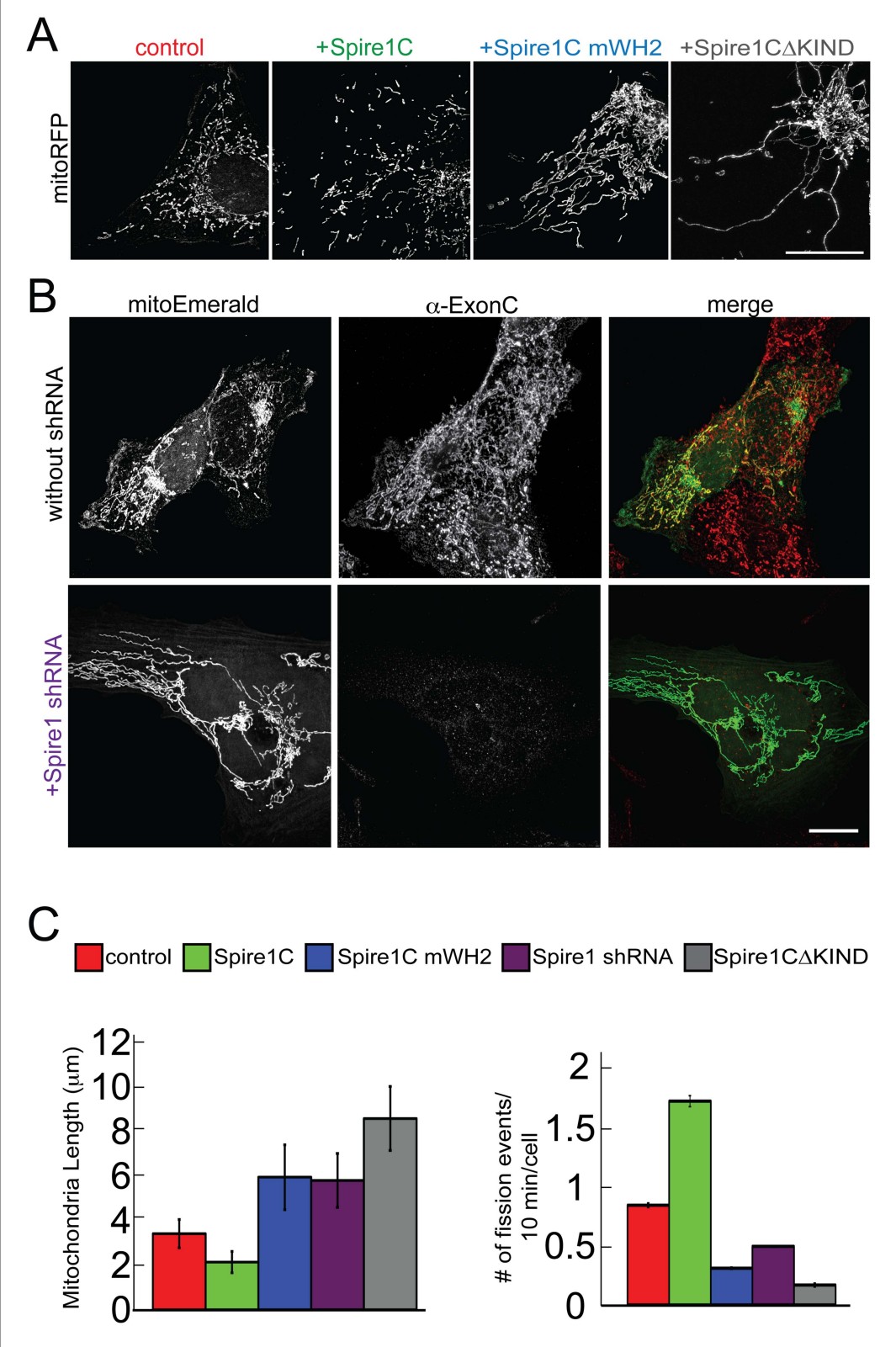

**Figure 4**. Spire1C promotes mitochondrial fission via its formin-binding KIND and actin-nucleating WH2 domains. (**A**) U2OS cells overexpressing Spire1C display shorter mitochondria (second panel, 2.2 ± 0.5 µm, $n_{mitochondria}$ = 211, $n_{cells}$ = 14, p < 0.0001), whereas cells overexpressing Spire1C mWH2 (third panel, 6.2 ± 1.52 µm, $n_{mitochondria}$ = 332,

*Figure 4. continued on next page*

*Figure 4. Continued*

$n_{cells}$ = 15, p < 0.0001) or Spire1CΔKIND (fourth panel, 9.0 ± 1.50 μm, $n_{mitochondria}$ = 232, $n_{cells}$ = 16, p < 0.0001) display longer, more tubulated mitochondria compared to control cells (first panel, 3.57 ± 0.45 μm, $n_{mitochondria}$ = 322, $n_{cells}$ = 34). Scale bar: 10 μm. (**B**) Cells transfected with mitoEmerald (and neighboring non-transfected cells) stained with α-ExonC (upper row) showed robust colocalization of mitoEmerald and α-ExonC, with a mixture of tubulated and fragmented mitochondria. Cells cotransfected with Spire1 shRNA and mitoEmerald with no detectable α-ExonC labeling (lower row) display long, tubulated mitochondria (6.1 ± 1.26 μm, $n_{mitochondria}$ = 222, $n_{cells}$ = 17). All primary antibodies were counterstained with Alexa-568 secondary antibody. Scale bar: 15 μm. (**C**) Left: Average mitochondrial lengths for control cells and cells overexpressing Spire1C, Spire1C mWH2, Spire1 shRNA or Spire1CΔKIND. Right: Average number of mitochondrial fission events in one cell in a timespan of 10 min for control ($n_{cells}$ = 17), Spire1C overexpressing ($n_{cells}$ = 10, p < 0.0001), Spire1C mWH2 overexpressing ($n_{cells}$ = 12, p < 0.0001), Spire1 knockdown ($n_{cells}$ = 25, p < 0.0001) and Spire1CΔKIND overexpressing ($n_{cells}$ = 22, p < 0.0001) cells. At least 3 separate experiments were performed for all conditions. Error bars represent standard error of the mean.

The following figure supplement is available for figure 4:

**Figure supplement 1**. Distribution of mitochondrial lengths measured in each condition.

we observed a significant decrease in ER-mitochondria overlap, as well as a decrease in the number of ER tubules crossing over mitochondria (*Figure 5A,B*). This suggested that the KIND domain of Spire1C might play a role in regulating the extent of ER-mitochondria intersections within cells, and that Spire1CΔKIND-mediated disruption of mitochondrial fission (see *Figure 4C*, right panel) could be due to a reduction in ER-mediated mitochondrial constriction in these cells (*Friedman et al., 2011*; *Korobova et al., 2013*; *Murley et al., 2013*).

## The Spire1C KIND domain directly interacts with ER-anchored INF2 to promote mitochondrial fission

One way the KIND domain of Spire1C could affect ER-mediated mitochondrial division is by binding to ER-anchored INF2. To test this possibility, we performed in vitro GST pull-down assays. We found that the C-terminal half of INF2 (INF2-CT), but not the N-terminal half (INF2-NT), associated with a GST-tagged Spire1C KIND domain, but not GST alone (*Figure 6A*). Addition of the N-terminal half of INF2 (INF2-NT) inhibited the interaction between Spire1C KIND and INF2-CT in our GST pulldown assays (*Figure 6A*; last well), suggesting that INF2's ability to self-interact (*Chhabra and Higgs, 2006*; *Ramabhadran et al., 2012*, *2013*) can regulate its association with Spire1C. We further confirmed the interaction between Spire1C's KIND domain and INF2 using fluorescence anisotropy (*Ramabhadran et al., 2013*) (*Figure 6B*), which showed a specific interaction between Spire1C's KIND domain and INF2. Taken together, these data demonstrate that the Spire1C KIND domain directly binds to INF2.

We next tested whether disrupting Spire1C's interaction with INF2 inhibits mitochondrial fission. To test this hypothesis, we first asked whether removing the KIND domain from Spire1C blocks the increase in mitochondrial division associated with overexpressing a constitutively active INF2 mutant, INF2 A149 (*Korobova et al., 2013*, *2014*). Consistent with previous reports (*Korobova et al., 2013*, *2014*), we found that overexpressing INF2 A149 resulted in significant shortening of mitochondria (*Figure 6C,E*; A149 alone). Similarly, co-overexpression of INF2 A149 and Spire1C or INF2 A149 and Spire1C mWH2 resulted in very short, fragmented mitochondria (*Figure 6C,E* +Spire1C or +Spire1C mWH2). By contrast, overexpression of INF2 A149 and Spire1CΔKIND significantly disrupted INF2 A149-mediated mitochondrial fragmentation (*Figure 6C*, +Spire1CΔKIND). These results suggest that while Spire1C actin-nucleating activity may be unnecessary for mitochondrial fission when INF2 is constitutively active, INF2 binding to the Spire1C KIND domain is necessary for INF2 to maximally induce mitochondrial fission. In further confirmation of the hypothesis that Spire1C and INF2 jointly work to drive mitochondrial fission, we found that siRNA-mediated knockdown of INF2 disrupted Spire1C-mediated upregulation of mitochondrial fission (*Figure 6D*). Taken together, the results suggest that Spire1C interacts with INF2 via Spire1C's KIND domain, and that this interaction promotes mitochondrial fission.

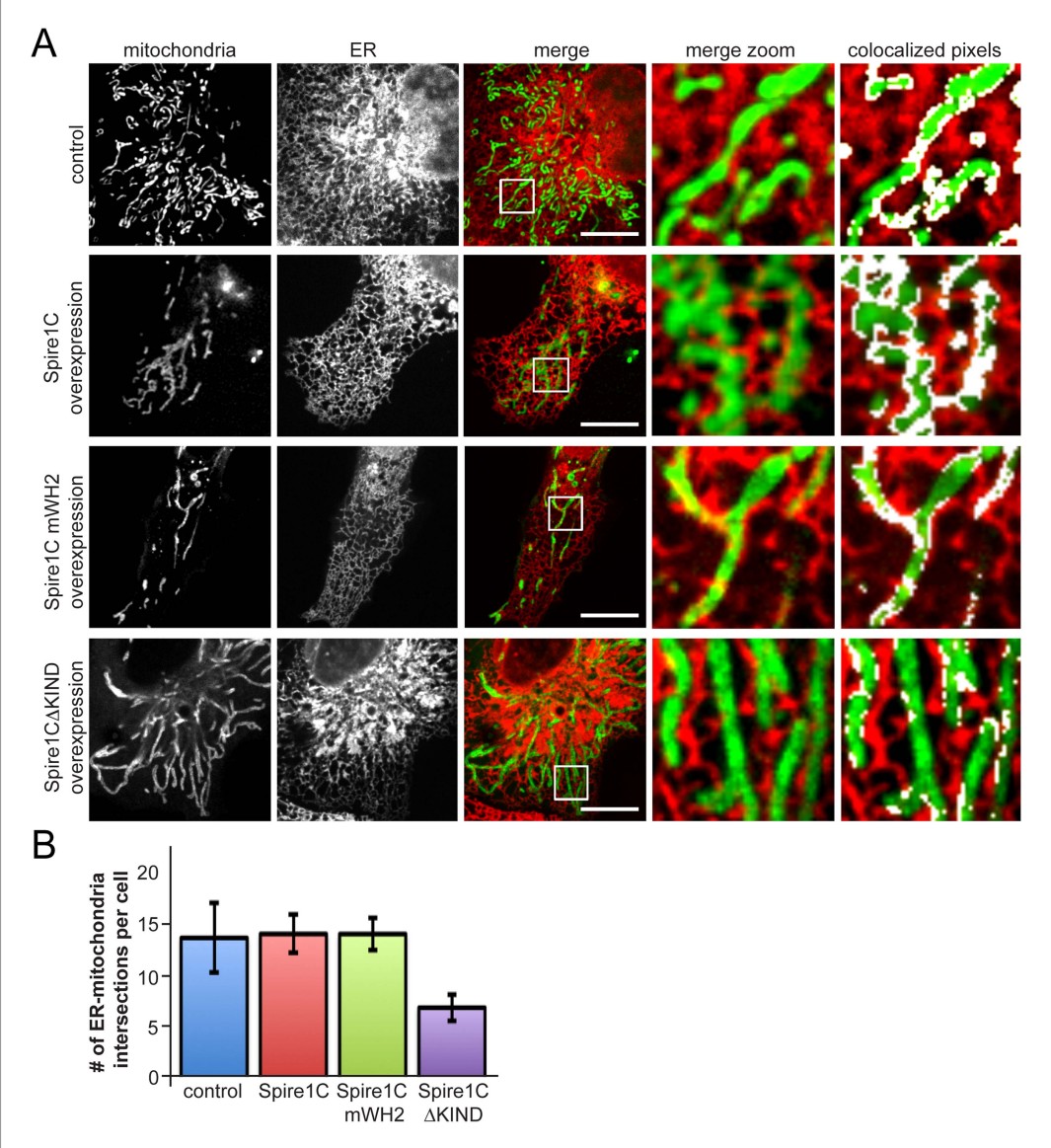

**Figure 5**. Spire1CΔKIND overexpression reduces the amount of ER-mitochondria overlap. (**A**) Confocal images of cells expressing Ii33-mCherry and overexpressing GFP-Spire1C (second row) or GFP-Spire1C mWH2 (third row), but not GFP-Spire1CΔKIND (fourth row), display significant overlap of mitochondria with ER, similar to control cells expressing mitoEmerald. The images on the right-hand side show a magnified view of the boxed region in the merge image, with overlapping pixels in displayed in white. Scale bar: 15 μm. (**B**) Bar graph representing the average number of ER-mitochondria intersections per cell. We were able to resolve an average of $14.3 \pm 3.5$ intersections in control cells ($n_{cells} = 12$). GFP-Spire1C overexpressing cells had $14.7 \pm 1.93$ ($n_{cells} = 15$) ER-mitochondria intersections per cell. GFP-Spire1C mWH2 expressing cells had an average of $14.7 \pm 1.62$ ($n_{cells} = 11$) ER-mitochondria intersections per cell. Spire1CΔKIND expressing cells had $6.8 \pm 1.33$ ($n_{cells} = 14$, $p < 0.05$) ER-mitochondria intersections per cell.

## Spire1C promotes ER-mediated mitochondrial constriction in an INF2-dependent fashion

Mitochondrial fission would be co-dependent on Spire1C and INF2 if Spire1C's interaction with INF2 drives ER-mediated mitochondrial constriction. To test this hypothesis, we employed confocal fluorescence imaging of ER and mitochondria to examine mitochondrial constriction sites in cells co-expressing the ER marker Ii33-mCherry and different variants of Spire1C. Mitochondria constriction

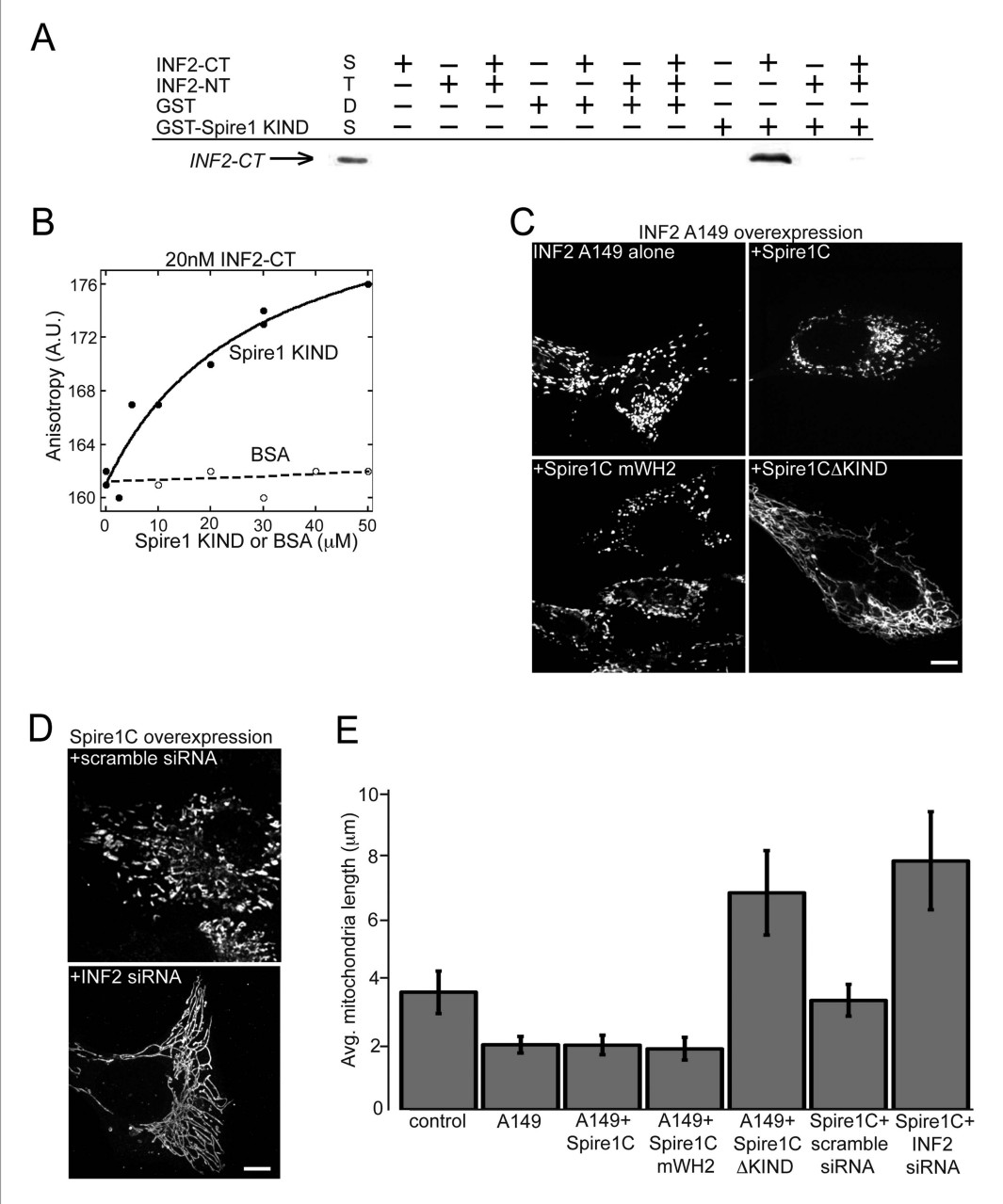

**Figure 6**. Spire1C and inverted formin 2 (INF2) directly interact and work together to regulate mitochondrial fission. (**A**) INF2-CT directly binds to Spire1-KIND in vitro. GST pull-down assays in actin polymerization buffer, containing combinations of the following: 20 µM GST or GST-Spire1 KIND bound to glutathione-sepharose beads; 1 µM INF2-CT; and 10 µM INF2-NT. Co-incubation of the GST-Spire1 KIND domain pulls down INF2-CT (third to last lane), but not INF2-NT (second to last lane). INF2-CT pulldown is inhibited by the addition of the INF2-NT (last lane). STDS lane represents 0.2 µM INF2-CT. (**B**) Fluorescence anisotropy binding curve of purified INF2-CT (20 nM) labeled with tetramethylrhodamine succinimide mixed with varying concentrations of Spire1 KIND or bovine serum albumin reveals a direct interaction between Spire1 KIND and INF2-CT. (**C**) Cells overexpressing a constitutively active INF2 mutant (INF2 A149 alone, $n_{cells}$ = 16, $n_{mitochondria}$ = 232) display very short, fragmented mitochondria compared to control cells (p < 0.0001). Cells overexpressing A149 and Spire1C (+Spire1C, $n_{cells}$ = 14, $n_{mitochondria}$ = 461) or Spire1C mWH2 (+Spire1C mWH2, $n_{cells}$ = 20, $n_{mitochondria}$ = 377) similarly display very short mitochondria. In contrast, cells overexpressing A149 and Spire1CΔKIND display longer, more tubulated mitochondria (+Spire1CΔKIND, $n_{cells}$ = 18, $n_{mitochondria}$ = 379, p < 0.0001). (**D**) Cells overexpressing Spire1C and treated with scrambled siRNA display shorter, more fragmented mitochondria (+scramble siRNA, $n_{cells}$ = 20, $n_{mitochondria}$ = 434, p < 0.05).
*Figure 6. continued on next page*

*Figure 6. Continued*

In contrast, cells overexpressing Spire1C and treated with INF2 siRNA display significantly longer, more tubulated mitochondria (Spire1C + INF2 siRNA, $n_{cells} = 22$, $n_{mitochondria} = 627$, $p < 0.0001$). Scale bars: 5 µm. (**E**) Bar graph displaying average mitochondria lengths for each of the conditions in this figure. Error bars represent standard error of the mean.

___

was visible at sites of ER-mitochondria crossover slightly more frequently in cells overexpressing Spire1C compared to cells not overexpressing the construct (*Figure 7A,B*, Spire1C, see arrows for ER-mediated mitochondrial constrictions, and arrowheads for ER-mitochondria intersections that didn't result in constrictions). Notably, cells overexpressing Spire1C mWH2 displayed significantly fewer mitochondrial constrictions at ER-mitochondria intersections (*Figure 7A,B*, Spire1C mWH2). Similarly, cells overexpressing Spire1CΔKIND showed a decrease in the frequency of constrictions at ER-mitochondria intersections (*Figure 7A,B*, Spire1CΔKIND). RNA-mediated Spire1C knockdown also resulted in decreased constrictions (*Figure 7A,B*, Spire1C shRNA). Finally, overexpressing Spire1C in cells treated with INF2 siRNA showed a significant reduction in mitochondrial constrictions (*Figure 7A, B*, INF2 siRNA + Spire1C). Taken together, our results suggest that Spire1C and INF2 work together to promote mitochondrial division by driving ER-mediated mitochondrial constriction, and that this process is dependent on Spire1C's ability to nucleate actin filaments on mitochondrial surfaces, as well as the ability for Spire1C and INF2 to interact via the Spire1C KIND domain.

## Simulations indicate that pressure from actin polymerization and actomyosin contraction forces are sufficient for driving mitochondrial constriction

Given the above experimental results, we used in silico simulations to test a potential model in which forces mediated by the actin cytoskeleton induce mitochondrial constriction. In this scenario, actin filament polymerization within the gap between the ER tubule surrounding the mitochondria and the mitochondrial outer membrane (*Figure 8A*) results in localized pressure that drives mitochondrial constriction to diameters required for Drp1 helix formation (*Korobova et al., 2013*). This pressure could originate either from forces exerted by actin polymerization against the mitochondrial outer membrane (*Korobova et al., 2013*), or by myosin-II dimer mediated contraction of actin filaments lying between the ER and mitochondrial membranes (*Hatch et al., 2014*; *Korobova et al., 2014*), or by a concerted action of these two complementary mechanisms.

To substantiate this, we created a simulation of mitochondrial constriction in response to a localized pressure generated by the above-mentioned mechanisms (*Figure 8B*, *Figure 8—figure supplement 1*, and *Figure 8—source data 1*). We modeled the constriction site of the mitochondrial outer membrane as a membrane tubule whose resistance to deformations is characterized by a bending modulus of $8 \times 10^{-20}$ Joules, typical for a lipid bilayer (*Helfrich, 1973*). The pressure deforming the membrane tubule was applied in the middle of the constriction zone along a strip of 50-nm thickness, corresponding to that of a typical ER tubule, while the computed shapes of the mitochondria constriction region corresponded to those of three different constriction events imaged with electron tomography (*Friedman et al., 2011*) (*Figure 8B*). The computed pressure values required for generation of these 3 degrees of mitochondrial constriction (*Figure 8—figure supplement 1* and *Figure 8—source data 1*) enabled us to calculate the numbers of polymerizing actin filaments, $N_f$, or the tension, $\gamma_m$, which has to be developed within the actin contractile system. Assuming that the force developed by one polymerizing actin filament is about 1 pN (*Footer et al., 2007*), the estimated filament number, $N_f$, varies between 10–20. The actomyosin tension values, $\gamma_m$, range from 2 to 3 pN. The obtained estimations for both $N_f$ and $\gamma_m$ are perfectly reasonable physiologically, which supports the feasibility of the suggested mechanisms. Thus, our results and model are fully consistent with previous studies suggesting that tightly regulated actin assembly at ER-mitochondria intersection sites facilitates mitochondrial membrane scission by Drp1 (*Friedman et al., 2011*; *Korobova et al., 2013*, *2014*; *Murley et al., 2013*; *Hatch et al., 2014*; *Li et al., 2015*).

## Discussion

A key event in the mitochondrion's life cycle is its division into distinct mitochondrial elements. Prior work studying this process demonstrated that division occurs at sites where ER wraps around

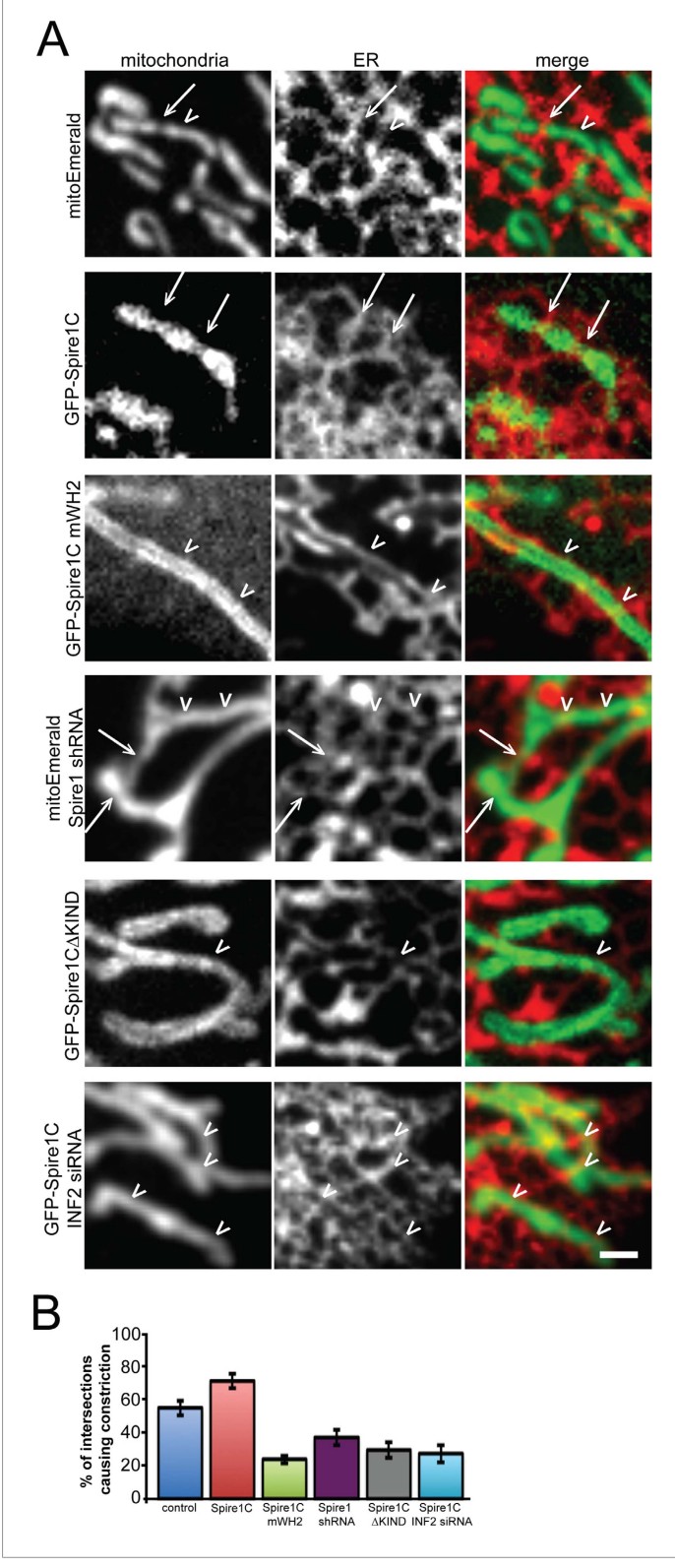

**Figure 7**. Spire1C overexpression enhances mitochondrial constriction via its WH2 and KIND domains in cooperation with INF2. (**A**) Representative confocal images of U2OS cells expressing Ii33-mCherry in order to visualize ER tubules crossing over mitochondria in cells expressing mitoEmerald (first row) or overexpressing GFP-Spire1C (second row), GFP-Spire1C mWH2 (third row), GFP-Spire1CΔKIND (fourth row), or GFP-Spire1C while
*Figure 7. continued on next page*

*Figure 7. Continued*

treated with INF2 siRNA (fifth row). Arrows indicate ER-mitochondria intersection points associated with mitochondrial constriction. Arrowheads indicate ER-mitochondria intersections not resulting in mitochondrial constriction. Scale bar: 1 μm. (**B**) Bar graphs representing the average percentage of ER-mitochondria intersections associated with mitochondrial constriction for each construct used. In cells expressing mitoEmerald, $55.2 \pm 5.5\%$ ($n_{intersections} = 172$, $n_{cells} = 12$) of ER-mitochondria intersections appeared to result in mitochondrial constriction. In GFP-Spire1C overexpressing cells, $71.5 \pm 4.5\%$ ($n_{intersections} = 221$, $n_{cells} = 15$, $p < 0.05$) of ER-mitochondria intersections resulted in mitochondrial constriction. In GFP-Spire1C mWH2 overexpressing cells, $24.1 \pm 2.4\%$ ($n_{intersections} = 162$, $n_{cells} = 11$, $p < 0.01$) of ER-mitochondria intersections appeared to result in mitochondrial constriction. In Spire1C knockdown cells, $37.2 \pm 5.7\%$ ($n_{intersections} = 123$, $n_{cells} = 11$, $p < 0.001$) of ER-mitochondria intersections appeared to result in mitochondrial constriction. In GFP-Spire1CΔKIND overexpressing cells, $29.5 \pm 4.7\%$ ($n_{intersections} = 95$, $n_{cells} = 14$, $p < 0.01$) of ER-mitochondria intersections appeared to result in mitochondrial constriction. In GFP-Spire1C overexpressing cells treated with INF2 siRNA, $27.5 \pm 5.3\%$ ($n_{intersections} = 178$, $n_{cells} = 16$, $p < 0.01$) of ER-mitochondria intersections appeared to result in mitochondrial constriction.

mitochondria (*Friedman et al., 2011*; *Murley et al., 2013*), with the ER providing a platform for actin polymerization mediated by the ER-anchored formin INF2 (*Korobova et al., 2013*, *2014*; *Hatch et al., 2014*). This actin meshwork is proposed to provide the force that drives mitochondrial constriction prior to Drp1-mediated mitochondrial division. Missing from this picture has been a molecular player that regulates INF2-mediated actin polymerization, ensuring that polymerization occurs specifically at ER-mitochondria contact sites to drive mitochondrial constriction and division. Here, we demonstrate that Spire1C, a novel mitochondrial outer membrane protein, can serve this role by both binding to INF2 as well as by acting as an actin-nucleator.

Spire proteins are membrane-binding actin-nucleators that interact with and regulate formin proteins (*Bosch et al., 2007*; *Quinlan et al., 2007*; *Pechlivanis et al., 2009*; *Pfender et al., 2011*; *Schuh, 2011*; *Vizcarra et al., 2011*; *Quinlan, 2013*). Given this, Spire proteins are potential candidates for regulating the actin polymerization activity of INF2 on mitochondrial membranes. In searching for such a protein, we identified a specific isoform of Spire1, Spire1C, which resides on mitochondria and interacts with INF2. Spire1C is distinct from other Spire proteins in having mitochondrial outer membrane localization. This localization is a result of Spire1C's unique ExonC domain, which serves as a mitochondria-targeting sequence. Spire1C undergoes lateral diffusion on the mitochondrial outer membrane, and is oriented with its formin-binding and actin-nucleating domains facing the cytoplasm. Spire1C promotes actin assembly on mitochondrial outer membranes; when Spire1C is overexpressed a massive buildup of actin around mitochondria is observed. The actin buildup is dependent on Spire1C's actin-nucleating WH2 domain, but not its formin-binding KIND domain. Therefore, Spire1C's canonical actin-nucleating domain drives actin accumulation on mitochondria independently of its interactions with formin proteins.

Given Spire1C's ability to assemble actin filaments on mitochondrial membranes, we examined whether modulating Spire1C's activity could affect mitochondrial lengths or their division rates, and we found that it can. Overexpressing Spire1C increases mitochondrial division rates while depleting Spire1C has the opposite effect, causing mitochondria to become highly elongated. Because the increased fission seen in cells overexpressing Spire1C depends not only on Spire1C's actin-nucleating WH2 domain but also its formin-binding KIND domain, we reasoned that Spire1C-mediated mitochondrial fission also depended on formin proteins. Given INF2's previously established role as a formin protein involved in ER-mediated mitochondrial constriction and division (*Korobova et al., 2013*, *2014*; *Hatch et al., 2014*), we hypothesized that Spire1C could be working together with INF2. We postulated that Spire1C could be promoting mitochondrial division by interacting with ER-anchored INF2, in order to enable mitochondria to come into close proximity with the ER so that actin-nucleation by Spire1C could enhance actin assembly mediated by INF2. Supporting this possibility, we found in GST pulldown and fluorescence anisotropy assays that the Spire1C KIND domain directly binds to INF2. In cells overexpressing Spire1C lacking its KIND domain-mediated INF2-binding activity (Spire1CΔKIND) or its actin-nucleating activity (Spire1C mWH2), ER-mitochondria associations leading to mitochondrial constrictions were significantly decreased. Moreover, overexpressing Spire1CΔKIND prevented mitochondria from dividing in cells expressing a constitutively active INF2 mutant (INF2 A149)

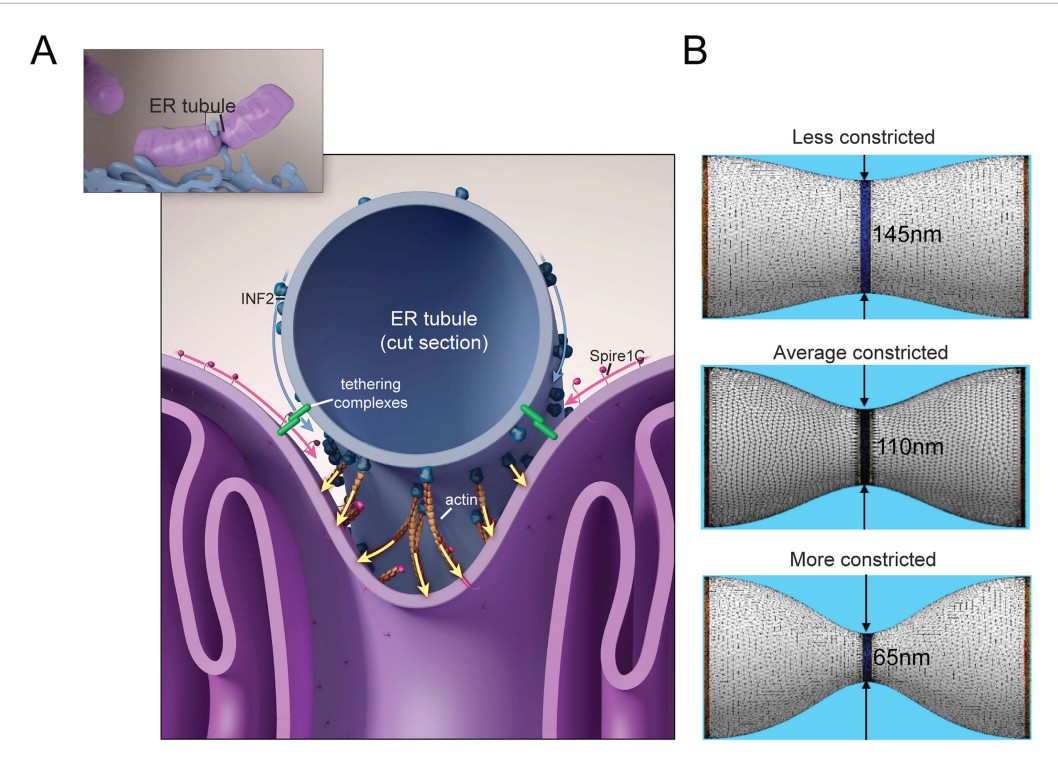

**Figure 8**. Putative model for how mitochondrial Spire1C and ER-anchored INF2 could mediate mitochondrial constriction via actin filament assembly. (**A**) Spire1C:actin complexes on mitochondria associate with INF2 on the ER. Actin filaments nucleated by Spire1C are elongated by the actin polymerization activity of INF2. The actin filament elongation activity exerts pressure on the mitochondrial outer membrane, thereby driving constriction of the latter. Tethering complexes may play a role in maintaining association between ER and mitochondrial membranes. Myosin-II dimers and the related contractile actin ring, which may also be involved in mitochondrial constriction, are not shown for simplicity. (**B**) Computational results showing mitochondrial shapes resulting from deformation by constricting pressure $P$ developed by the actin polymerization and/or actin contractile based mechanisms (see also *Figure 8—figure supplement 1*, *Figure 8—source data 1*, and 'Materials and methods' for more information). The mitochondrial constriction site was modeled as a tubular membrane of about 680 nm length and with initial radius $R = 230$ nm. The dark blue strip in the middle represents the 50 nm wide zone of the pressure application. The images correspond to 3° of the mitochondria constriction characterized by cross-sectional radii $r$ in the narrowest place of 145 nm, 110 nm and 65 nm. The corresponding values of the pressure $P$, the required numbers of the polymerizing actin filaments, $N_f$, and the required tensions in the actin contractile ring, $\gamma_m$, are presented in *Figure 8—figure supplement 1* and *Figure 8—source data 1*.

The following source data and figure supplement are available for figure 8:

**Source data 1**. Specific values of the system parameters and the computational results for the three specific extents of mitochondrial constriction presented in *Figure 8*, *Figure 8—figure supplement 1*, and discussed in the main text.

**Figure supplement 1**. Computational results of simulations of mitochondrial constriction mediated by actin polymerization and actin constriction mechanisms.

that normally induces dramatic mitochondrial fission (*Korobova et al., 2013*, *2014*). Finally, Spire1C overexpression in cells lacking INF2 failed to induce mitochondrial fission.

All these observations suggest a model in which mitochondrial Spire1C and ER-anchored INF2 conspire to mediate mitochondrial constriction via actin filament assembly. In this scheme, Spire1C: actin complexes on mitochondria associate with INF2 on the ER, acting together with other ER-mitochondria tethering complexes (*Rowland and Voeltz, 2012*) to draw the two organelles together. This results in the ER wrapping around the mitochondria. Once this occurs, actin filaments

nucleated by Spire1C are elongated by the actin polymerization activity of INF2, similar to the 'rocket launcher' mechanism shown for other actin nucleating and formin proteins (*Breitsprecher et al., 2012*). Because INF2 can both polymerize and sever actin filaments, a complex meshwork of actin grows between the ER and mitochondria, which may be further impacted by myosin-II dimer recruitment (*Hatch et al., 2014*; *Korobova et al., 2014*) as well as perhaps other actin-regulatory proteins such as cofilin or Arp2/3 (*Derivery et al., 2009*; *Liu et al., 2009*; *Li et al., 2015*). The growing actin meshwork between the ER and mitochondria then exerts pressure on the mitochondrial outer membrane causing its constriction. Our computational modeling of this process confirmed that polymerizing actin filaments from this meshwork has sufficient force to bend and constrict the mitochondrial membrane once the filaments abut the mitochondria surface.

Clearly, further work is needed to clarify the mechanism by which Spire1C and INF2 facilitate mitochondrial division. First, a better understanding of how Spire1C and INF2 interact with and regulate one another's activities during mitochondrial constriction is required. The relatively low affinity between Spire1C's KIND domain and INF2 detected in our assays, if applicable to living cells, is consistent with Spire1C:INF2 dissociation once INF2 begins to elongate actin filaments. Second, as several proteins are known to tether ER-mitochondrial membranes (*Rowland and Voeltz, 2012*), the role of these proteins in promoting or disrupting Spire1C's interaction with INF2 also needs to be studied. Such interactions may underlie differences in the mode of mitochondrial division seen in cells undergoing apoptosis, mitophagy, mitosis, or in response to toxins such as LLO from *Listeria* (*Chan, 2012*; *Hoppins and Nunnari, 2012*; *Youle and van der Bliek, 2012*; *Stavru et al., 2013*). Finally, the precise organization of the actin meshwork responsible for constricting mitochondria needs to be characterized at higher resolution. This will help determine whether the actin meshwork constricts mitochondria by myosin-mediated contraction, by elongating filaments pushing, or by a combination of both.

While we have focused on Spire1C's role in mitochondrial constriction, the establishment of Spire1C as a mitochondrial outer membrane protein suggests that Spire1C is optimally positioned to serve as a molecular hub that links mitochondrial dynamics to the actin cytoskeleton as well as to the ER. While our appreciation of the role of actin in mitochondrial division is rapidly growing (*De Vos et al., 2005*; *Korobova et al., 2013*, *2014*; *Hatch et al., 2014*; *Li et al., 2015*), there are other important functions for actin in mitochondrial dynamics, such as mitochondrial motility in neurons (*Hollenbeck and Saxton, 2005*; *Pathak et al., 2010*), mitochondrial partitioning prior to cell division in fibroblasts (*Quintero et al., 2009*; *Rohn et al., 2014*), and perhaps also the partitioning of mitochondrial DNA (*Boldogh et al., 2003*, *2004*; *Reyes et al., 2011*). This list is almost certainly not exhaustive; there may yet be other known and unknown roles for the actin cytoskeleton in mitochondrial biology, and vice versa. Interestingly, Spire1C directly interacts with the tail domain of myosin Va (data not shown), an actin-binding motor protein that has been shown to be involved in both mitochondrial and ER movement in neurons (*Wagner et al., 2011*). In other cellular systems myosin Vb, Rab11a, and Spire proteins cooperate to drive actin-based vesicle movements and dynamics (*Schuh, 2011*; *Montaville et al., 2014*)—perhaps similar mechanisms exist for mitochondrial movements. Along these lines, it is interesting to note that Rab11a has also been implicated in mitochondrial dynamics (*Landry et al., 2014*)—exploring these findings in the context of Spire1C function may provide new insight towards mitochondrial movements and dynamics, and perhaps the relationship between actin-dependent motility and actin-dependent fission. Finally, the recent discovery of a role for the ER in mediating endosomal constriction and division raises the possibility that endosomal isoforms of Spire (*Kerkhoff, 2006*; *Liu et al., 2009*) are playing a similar role in promoting ER/actin/INF2-mediated endosomal fission. In fact, results from previous studies suggest that overexpression of the endosomal Spire2 protein lacking its KIND domain may result in endosome elongation (*Dietrich et al., 2013*), which would be analogous to what we have observed for Spire1CΔKIND overexpression and mitochondria.

In conclusion, our identification and characterization of Spire1C as an ER- and actin-binding mitochondrial outer membrane protein opens the door for novel avenues towards understanding the regulation of myriad roles of actin, mitochondria, and the ER in cellular function and disease (*Rappold et al., 2014*).

## Materials and methods

### Cell culture and transfections

U2OS and Cos-7 cells were purchased from ATCC (Manassas, VA). U2OS cells stably expressing GFP-INF2 was described in *Chhabra et al. (2009)*. All cells were grown in DMEM (Invitrogen, Carlsbad, CA)

with 10% fetal bovine serum. For imaging, fibronectin coated coverslips ranging between 168 and 172 µm (for fixed cell imaging) or #1.5 LabTek chambers (for live cell imaging) were incubated with 10 µg/ml of fibronectin in PBS at 37°C for 30 min prior to plating the cells. Transient transfections were performed using FuGene 6 (Promega, Madison, WI) according to the manufacturer's recommendations. For overexpression experiments, 1 µg of DNA per coverslip was used. For minimal perturbation while imaging ER and mitochondria, 50 ng of DNA was used as described in *Friedman et al. (2011)*. For siRNA transfections, cells were treated as in *Korobova et al. (2013)*. Briefly, U2OS cells stably expressing GFP-INF2 (*Chhabra et al., 2009*) were plated on 6-well plates, treated with 63 pg of siRNA per well, and analyzed 72 hr post-transfection. siRNA or shRNA-mediated knockdown was confirmed by loss of GFP-INF2 or GFP-Spire1C fluorescence.

## Plasmids and siRNA oligonucleotides

Ii33-mCherry was a generous gift from P Satpute (National Institutes of Health, Bethesda, MD). MitoEmerald and mitoRFP were gifts from A Rambold (National Institutes of Health, Bethesda, MD). Spire1C was amplified from mouse brain cDNA using the sequence of NM_194355 as a reference. We expected to obtain a sequence yielding protein corresponding to GI 37595748, however, it contained an additional 58 residues (ExonC). Thorough examination of all available mammalian Spire1 isoforms revealed at least 3 alternatively-spliced exons, which we refer to as exons A (majority of KIND domain), B (protein sequence AVRPLSMSHSFDLS), and C (protein sequence VPRITGVWPRTPFRPLFSTIQT ASLLSSHPFEAAMFGVAGAMYYLFERAFTSRWKPSK).

To obtain a full-length Spire1C construct, we amplified the mouse Spire1 gene AK129296, which contains the full KIND domain through the first 3 WH2 domains, along with NM_194355, which contains a partial KIND domain and ExonC without Exon B. A series of amplifications of partial gene sequences was then performed to obtain versions of mouse Spire1 that were ± each of exons A, B, and C (*Figure 1—figure supplement 1*). Each variant of the Spire1 gene was cloned into the AscI and PacI sites of modified pCS2+ vectors containing epitope tag sequences adjacent to the multiple cloning site, creating Spire1C constructs with either N-terminal 6x-myc or fluorescent protein tags. For the FPP assay and knockdown experiments, human Spire1C ORF XM_005258122 (acquired from Genscript USA Inc., Piscataway Township, NJ) was cloned into the pEGFP-C1/pmApple-C1 and pEGFP-N1/pmApple-N1 vectors (Clontech) using the XhoI-BamHI and NheI-AgeI restriction sites, respectively. A nucleation-deficient mutant of Spire1C (Spire1C mWH2) was generated by utilizing internal PstI and AfeI restriction sites in the Spire1C gene. A sequence of 822 nucleotides of the Spire1C gene between internal PstI and AfeI sites was synthesized (Genscript USA Inc.) that contained alanines in place of the key hydrophobic residues required for nucleation in all four WH2 domains (*Quinlan et al., 2005*; *Loomis et al., 2006*; *Quinlan et al., 2007*). Insertion of the alanine-mutated WH2 domains was confirmed with DNA sequencing. All primers used are shown below along with the WH2 mutant insertion.

Primers for spire1 gene amplification and plasmid construction

Primer 1 GCGC<u>GGCGCGCC</u>ATGGAACTGCATACATTTCTGACCAAAATTAAGAG
Primer 2 GCGC<u>TTAATTAA</u>TCAGATCTCGTTGATAGTCCGTTCTGAAG
Primer 3 GAGCA<u>GGCGCGCC</u>ATGGCCAATACCGTGGAGGCTG
Primer 4 GGCG<u>TTAATTAA</u>TCTAGTCTGCTCCGTCTAATTTCTTC
Primer 5 GACG<u>GCGCGCC</u>ATGGCGCAGCCCTCCAG
Primer 6 GGCG<u>TTAATTAA</u>TCTAGTCTGCTCCGTCTAATTTCTTC
Primer 7 CCATGTGCTCCAGGAAGAAGCC
Primer 8 CTGCCTTCCAAGCCATACTCTACTCTAC

All primers are 5′ to 3′. AscI or PacI restriction sites are <u>underlined.</u>

## WH2 mutant insertion sequence

CGAGGCTGCAGATGAAGGCCCGGAAGATGAAGACGGAGAG
AAGAGAAGCATCTCAGCCATCCGGTCCTATCAGGACGTTATGAAG
ATCTGTGCTGCTCACCTCCCAACTGAGTCGGAGGCACCCAATCAT
TATCAGGCAGTATGTCGGGCCCTGTTCGCAGAAACCATGGAACTG
CATACATTTCTGACCAAAATTAAGAGTGCAAAGGAGAACCTTAAG
AAGATTCAAGAAATGGAAAAGGGTGATGAATCTAGCACAGATCTG
GAGGACCTGAAAAATGCAGACTGGGCCCGGTTCTGGGTACAAGCG

GCGAGGGATTTGCGAAATGGGGTAAAAGCTAAGAAAGTCCAGCAG
CGGCAGTACAACCCTCTGCCCATTGAGTACCAACTGACCCCTTAT
GAGATGGCCGCGGACGACATTCGGTGCAAAAGATACACCGCGAGA
AAAGTAATGGTAAATGTGACGTCCCCCCTCGGTTGAAAAAGAGT
GCTCATGAGGTCGCCGCTGACTTTATCAGATCAAGACCCCCTGCA
AATCCAGTTTCAGCCAGAAAACTGAAACCAACCCCACCACGGCCA
CGGAGCCTCCATGAAAGAGCAGCAGAAGAAATTAAAGCAGAAAGA
AAGGCTCGGCCTGTGTCACCAGAAGAAATTAGACGGAGCAGACTA
GCAGTGCGGCCACTTAGCATGTCTCACAGTTTTGACTTGTCAGAT
GTCACTACGCCAGAATCTCCAAAGAATGTTGGAGAATCATCTATG
GTGAATGGAGGCTTAACATCTCAAACAAAAGAAAATGGGCTGAGC
GCTGCCCAGCAGGGGTC

The entire amino acid sequence of Spire1C is below

MAQPSSPGGEGPQLGAAGGPRDA
LSLEEILRLYNQPINEEQAWAVCFQCCGSLRAAAARRQPHRRVRSAAQIRVWRDGAVTLAPAAAAAAE
GEPPPASGQLGYSHCTETEVIESLGIIIYKALDYGLKENEERELSPPLEQLIDQMANTVEADGSKDEGYEAAD
EGPEDEDGEKRSISAIRSYQDVMKICAAHLPTESEAPNHYQAVCRALFAETMEL
HTFLTKIKSAKENLKKIQEMEKGDESSTDLEDLKNA
DWARFWVQVMRDLRNGVKLKKV
QQRQYNPLPIEYQLTP
YEMLMDDIRCKRYTLRKV
MVNGDVPPRLK
KSAHEVILDFIRSRPPLNPV
SARKLKPTPPRPRS
LHERILEEIKAERKLRPV
**S**PEEIRRSRL
AVRPLSMSHSFDLS
DVTTPE**S**PKNVGESSMVNGGLTSQTKENGLSAAQQGSAQRKRLLKAPTLAELDS**S**D**S**EEEKSLHKSTSS
SSASPSLYEDPVLEAMCSRKKPPPKFLPIS**S**TPQPERRQPPQRRHSIEKETPTNVRQFLPPSRQSSRSL
VPRITGVWPRTPFRPLFSTIQTASLLSSHPFEAAMFGVAGAMYYLFERAFTSRWKPSK
EEFCYPVEC
LALTVEEVMHIRQVLVKAELE
KYQQYKDVYTA
LKKGKLCFCCRTRRFSFFTWSYTCQFCKRPVCSQCC
KKMRLPSKPYSTLPIFSLGPSALQRGESCSRSEKPSTSHHRPLRSIARFSTKSRSVDK**S**DEELQFPKELMED
WSTMEVCVDCKKFISEIISSSRRSLVLANKRARLKRKTQSFYMSSAGPSEYCPSERTINEI

KIND domain
WH2 domains (mutated to alanine to make nucleation deficient)
Alternate exon B
Alternate ExonC
Spire box
mFYVE domain

Spire1 shRNA constructs were generated by cloning the sequences into Clontech's pSingle-tTS-shRNA vectors using the HindIII/XhoI restriction sites. The sequences cloned into the vector to knockdown Spire1C were 5′-TCGAGGGATTAGACGTAGCAGATTATTCAAGAG ATAATCTGCTACGTCTAATCT TTTTTACGCGTA-3′ (Spire1C 3′ UTR, used for fixed cell imaging) and 5′-TCGAGGCGAATAATCTC CTGACTAATTCAAGAGATTAGTCAGGAGATTATTCGTTTTTTACGCGTA-3′ (Spire1C ORF, used for live cell imaging). Oligonucleotides for human INF2 siRNA were previously described in *Korobova et al. (2013)*. Briefly, the sequence used to knockdown INF2 was 5′-ACAAAGAAACTGTGTGTGA-3′, and as a control, Silencer Negative Control #1 (Ambion) was used.

## Amplification of alternate ExonC DNA from mouse tissue panel

Oligos flanking ExonC were designed to amplify the *spire1C* gene. A mouse tissue cDNA panel (Clontech, Mountain View, CA) was used as a template for amplification using Primers 7 and 8.

Amplified DNA containing ExonC was ~400 bp, while DNA lacking this exon was ~200 bp. Multiple oligomer sets were utilized to eliminate non-specific amplification while capturing as many on-target amplifications as possible. PCR reactions were run on a 2% agaorse gel, and bands of the appropriate size were excised from the gel and purified using a QIAquick gel extraction kit (Qiagen, Germantown, MD). Purified DNA was cloned into the pCR II-TOPO vector using the TOPO-TA cloning kit (Invitrogen). Sequence analysis was used to confirm the sequence of the amplified and inserted DNA.

### Plasmids for antibody generation

The Spire1C gene was amplified from mouse cDNA as described above. Vectors used for protein purification include a modified avidin-6x his-MBP-TEV-3x FLAG-Precision construct under the P1 promoter, as well as a modified pGEX vector for N-terminal GST fusion proteins. All vector backbones were gifts of Dr. Aaron Straight and are described in *Figure 1—figure supplement 2*.

### Antibody production and affinity purification

ExonC and the C-terminal 50 residues of mouse Spire1 were cloned into the avi-his-MBP-TEV-3xFLAG-precision vector described above or a modified pGEX vector (for N-terminal GST fusion proteins) using standard techniques (*Figure 1—figure supplement 2*). Avi-his-MBP-TEV-3xFLAG-precision constructs were expressed and purified as described above with the following modifications. For avi-his-MBP-TEV-3xFLAG-precision constructs, protein was purified over Ni-NTA resin, and the eluate was further purified on an S-200 gel filtration column, followed by a HiTrap Q column to remove any contaminating DNA. Protein samples were sent to Cocalico Biologicals and injected into rabbits to produce antisera.

GST fusion proteins were used for affinity column construction for affinity purification of antibodies. These proteins were purified by using single colonies of transformed Rosetta (DE3) cells to inoculate 400 ml Terrific Broth (TB; Invitrogen) cultures containing 100 μg/ml carbenicillin, 34 μg/ml chloramphenicol, which was grown overnight at 37°C. This culture was diluted into 2 l of fresh TB with antibiotics and grown to an O.D. of 0.8–0.9, at which time it was moved to 23°C. After 1 hr at 23°C, cultures were induced with 0.5 mM isopropyl β-D-1-thiogalactopyranoside for 3–4 hr and harvested as described for purification of avi-his-MBP-TEV-3xFLAG-Precision proteins above. Cells were thawed in lysis buffer (50 mM Tris, 1 M NaCl, 1 mM EDTA, 1 mM DTT, and protease inhibitor cocktail, pH 7.8), sonicated, and lysates were centrifuged for 30 min at 125,000×*g* 4°C. Supernatant was applied to hydrated glutathione resin (2 ml bed volume per liter culture), protein bound for 1 hr at 4°C, and resin was washed extensively with lysis buffer. Protein was eluted with elution buffer (50 mM Tris, 150 mM NaCl, 1 mM EDTA, 20 mM reduced glutathione, 1 mM DTT, and protease inhibitor cocktail, pH 7.8) and loaded onto a HiTrapQ anion exchange column. A salt gradient of 150 mM to 1 M was used for protein elution.

### Affinity column construction

Spire1 affinity columns were made using GST-fusion proteins following the method of *Finan et al. (2011)*. Proteins were coupled to Affi-Gel 10 by washing with 5 resin/column vol (CV) of 0.2 M glycine, pH 2 and quickly equilibrating with PBS. Antisera was filtered through a 0.2 μm filter and flowed over the column continuously overnight. The column was washed with 50 CV wash buffer (PBS with 500 mM NaCl and 0.1% Tween 20), followed by 2.5 CV 0.2× PBS. Antibody was eluted with 1 CV 0.2 M glycine, pH 2 directly into 1 M Tris, pH 8.5 to neutralize the solution. Concentration of elution fractions was checked on a Nanodrop spectrophotometer using the IgG setting. The column was washed with 20 CV PBS, the antisera was re-filtered, and the purification process was repeated to isolate additional antibody. Fractions with an O.D. > 0.2 were pooled, dialyzed into PBS containing 50% glycerol, and stored at −20°C. Antisera to ExonC yielded no IgG after affinity purification. Instead, whole antisera were used for subsequent experiments to probe ExonC function and localization.

### Antibody characterization

Two rabbit polyclonal antibodies described above and three commercially available antibodies were used for examining expression patterns of Spire1 protein in various cell types. The antibodies discussed are: (1) Rabbit polyclonal anti-Spire1 C-term (affinity-purified), (2) Rabbit polyclonal anti-Spire1 ExonC (whole antisera), (3) Sigma mouse monoclonal anti-Spire1, (4) Abcam (Cambridge, UK) mouse monoclonal anti Spire1, and (5) Abnova (Taipei, Taiwan) rabbit antisera to Spire1. Notably, all of the

commercially-available antibodies were targeted to residues 482–584 of the Spire1 isoform lacking ExonC (NP_064533), and thus could only detect non-ExonC containing isoforms.

Western blots were performed with 5–50 µg cell lysate and antibodies/antisera was tested at various concentrations, temperatures, and lengths of time for best conditions. Optimized conditions for all antibodies used in this work are described below. HRP-conjugated goat anti-rabbit secondary antibody was used at 1:20,000 in all cases.

| Source | Species | Type | Target region | IB dilution | Time | Temperature | IF dilution |
|---|---|---|---|---|---|---|---|
| Sigma | mouse | monoclonal | 482–584 of NP_064533; 'last 100 a.a.'; flanks (but does not contain) z' | 1:2000 | 1 hr | 23°C | n/a |
| | | | | – | O/N | 4°C | n/a |
| Abcam | mouse | monoclonal | | 1:1000 | O/N | 4°C | n/a |
| Abnova | rabbit | antisera | | 1:1000 | O/N | 4°C | n/a |
| This study | rabbit | polyclonal, affinity purified | last 51 residues | 1:2000 | O/N | 4°C | 1 to 100 |
| | | | | 1:3000 | 1 hr | 23°C | – |
| This study | rabbit | antisera | ExonC | 1:1000 | O/N | 4°C | 1 to 500 |

## GST pulldown assay

Spire-KIND (amino acids 1–234) was expressed as a GST fusion protein in bacteria, and purified on glutathione-sepharose (GE Biosciences, Buckinghamshire, UK) followed by Superdex200 gel filtration (GE Biosciences) of the glutathione-eluted GST-fusion protein. GST-KIND or GST alone was re-bound to glutathione-sepharose in binding buffer (50 mM KCl, 1 mM MgCl$_2$, 1 mM EGTA, 10 mM Hepes-HCl pH 7.4, 1 mM DTT, 0.02% thesit (Sigma, St. Louis, MO), 10 µg/ml aprotinin, 2 µg/ml leupeptin, 0.5 mM benzamidine). INF2-CT (amino acids 469–1249, containing FH1, FH2 and C-terminal regions) and INF2-NT (amino acids 1–420, containing DID and dimerization region) were purified as described (*Ramabhadran et al., 2012*). Proteins were mixed at 20 µM GST protein, 1 µM INF2-CT and 10 µM INF2-NT in binding buffer and incubated overnight, then quickly washed once in binding buffer. Proteins in glutathione sepharose-bound pellet were resolved by SDS-PAGE.

## Anisotropy binding assay

INF2 C-term (human CAAX variant, amino acids 941–1249) was expressed in bacteria, purified and labeled on its N-terminal amine with tetramethylrhodamine succinimide as described (*Ramabhadran et al., 2013*). Labeled INF2-C-term (20 nM) was mixed with varying concentrations of Spire-KIND or bovine serum albumin (BSA) in 10 mM Hepes pH 7.4, 50 mM KCl, 1 mM MgCl$_2$, 1 mM EGTA, 1 mM DTT, 0.5 mM Thesit detergent (nonaethylene glycol monododecyl ether) at 23°C for 1 hr before reading fluorescence anisotropy in an M-1000 fluorescence plate reader (Tecan Inc) at 530 nm excitation and 585 nm emission.

## Immunofluorescence

Cells were washed in phosphate buffered saline (PBS; pH 7.4) then fixed with 4% paraformaldehyde for 30 min. Cells were then permeabilized with 0.1% Triton X-100 for 30 min before being blocked overnight with 4% BSA at 4°C. The next day, cells were incubated with primary antibody for 2 hr, rinsed three times with PBS for 10 min each, then incubated with secondary antibodies (Invitrogen) for 1 hr, rinsed three times with PBS for 10 min each, then counterstained with phalloidin (Invitrogen) for 30 min, then rinsed with PBS three times, then mounted using ProLong Gold antifade reagent.

## Confocal imaging

Confocal images were acquired with an Apochromat 63× 1.4 NA objective lens (Carl Zeiss, Jena, Germany) on a Marinas spinning disk confocal imaging system (Intelligent Imaging Innovations, Denver, CO) using an EM charge-coupled device camera (Evolve; Photometrics, Tucson, AZ), or a 100× Apo TIRF 1.49 NA objective (Nikon Instruments, Tokyo, Japan) on a Yokogawa CSU-X1 spinning disk system using an EM charge-coupled device camera (Evolve; Photometrics). Cells were imaged in HEPES-buffered growth media. Confocal z-stacks were taken using 200 nm steps. Images were deconvoluted using Slidebook 6. Individual 16-bit tiff image files were exported, then processed using ImageJ.

## Photobleaching of GFP-Spire1C in specific cellular regions

GFP-Spire1C photobleaching experiments presented in *Figure 2* and *Figure 2—figure supplement 1* were carried out on a Marianas spinning disc confocal microscope equipped with a Mosaic Digital Illumination System. The laser power entering the Mosaic was 9 mW. Image acquisition and photobleaching of GFP-Spire1C (including region selection and 405 laser exposure control) were carried out using Slidebook 6 software.

## SIM imaging

SIM imaging of fixed cells was performed using an ELYRA SIM (Carl Zeiss) with an Apochromat 63×1.4 NA oil objective lens. Five angles of the excitation grid with five phases each were acquired for each channel and each z-plane, which were spaced at 110 nm each. SIM processing was performed using the SIM module of the Zeiss Zen software package. 16-bit grayscale tiffs were subsequently exported to ImageJ for quantification and processing into rendered colored images. Channels in maximum projection images were aligned in the xy-plane using maximum projection images of fluorescent beads.

## Image processing and analysis

All image analysis and processing was performed using ImageJ. Mitochondria lengths were measured manually by first setting the scale according to pixel size, drawing a line along the length of the mitochondria, then using ImageJ's 'measure' function. Colocalization analysis and rendering was performed using the colocalization plugin included in the MacBiophotonics ImageJ plugin bundle (http://rsb.info.nih.gov/ij/plugins/mbf/index.html). When calculating Pearson's values, the mitochondria channel was used as a mask for colocalization. ER-mitochondria intersection sites were visually identified as regions where ER tubules could be clearly visualized crossing mitochondria—these regions were always in the periphery of the cell, significantly restricting the total number of intersections that could be reliably identified. Mitochondria constriction sites were visually identified as regions defined by relative narrowing of mitochondria diameter or reduced fluorescence. Magnifications of boxed regions were generated using ImageJ. Color images of merged 16-bit tiffs exported from the microscope were generated using the ImageJ 'merge channels' function. Statistical analysis was performed using Excel (Microsoft, Redmond, Washington). p-values were determined using the unpaired Student's *t*-test or ANOVA, as appropriate.

## Modeling

The deformed shapes of the membrane tubule representing the constriction site of the mitochondrial outer membrane were determined by minimizing the energy of the membrane bending upon the condition of a given pressure $P$ acting on a limited region in the middle of the tube (*Figure 8B*).

The value of the bending energy, $F_B$, was determined by

$$F_B = \int \frac{1}{2} \kappa\, J^2\, dA, \tag{1}$$

where $\kappa = 8 \times 10^{-20}$ Joule is the lipid bilayer bending modulus, $J$ is the local total curvature of the membrane surface changing along the membrane surface and equal at each point to the sum of the local principal curvatures (*Helfrich, 1973*; *Spivak, 1979*). The integration in *Equation 1* is performed over the whole surface of the deformed tubule.

The boundary conditions for the energy minimization consisted in the requirements that at the tubule left and right edges (i) the tubule cross-sectional radius, $r$, remains equal to its initial (preceding the deformation) value $r = R$, and (ii) the tubule profile remains parallel to the tubule axis. While the tubule length $L = 680$ nm was required to remain constant during the deformation, the tubule surface area was free to change. This means that the membrane lateral tension was taken to be zero, which guaranteed that the membrane bending energy was the sole contribution to the membrane elastic energy.

The energy minimization and the shape determination for each pressure value were performed using Brakke's 'Surface Evolver' program (*Brakke, 1992*).

## Acknowledgements

We would like to thank Margot Quinlan for purified Spire1C-KIND protein and for helpful suggestions. Richard Youle, Chunxin Wang, John Hammer, and the members of the Lippincott-Schwartz lab all provided helpful comments and suggestions on the manuscript.

## Additional information

### Competing interests

MK: Reviewing editor, *eLife*. The other authors declare that no competing interests exist.

### Funding

| Funder | Grant reference | Author |
| --- | --- | --- |
| Israel Science Foundation (ISF) | 758/11 | Gonen Golani, Michael Kozlov |
| National Institutes of Health (NIH) | 5R01GM106000-03 | Henry Higgs |
| National Institutes of Health (NIH) | 5R01GM033289-32 | James Spudich |
| National Institutes of Health (NIH) | 1ZIAHD001609-21 | Uri Manor, Jennifer Lippincott-Schwartz |
| National Institutes of Health (NIH) | 1ZIAHD008928-02 | Eric Christenson |

The funders had no role in study design, data collection and interpretation, or the decision to submit the work for publication.

### Author contributions

UM, SB, Conception and design, Acquisition of data, Analysis and interpretation of data, Drafting or revising the article, Contributed unpublished essential data or reagents; GG, Acquisition of data, Analysis and interpretation of data; EC, Acquisition of data, Analysis and interpretation of data, Contributed unpublished essential data or reagents; MK, Conception and design, Acquisition of data, Analysis and interpretation of data; HH, Conception and design, Acquisition of data, Analysis and interpretation of data, Drafting or revising the article; JS, Conception and design, Analysis and interpretation of data, Contributed unpublished essential data or reagents; JL-S, Conception and design, Analysis and interpretation of data, Drafting or revising the article

### Author ORCIDs

Uri Manor, http://orcid.org/0000-0002-9802-1955

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
