## [Decision Letter]

Thank you for submitting your work entitled “A mitochondria-anchored isoform of the actin-nucleating Spire protein regulates mitochondrial division” for peer review at *eLife*. Your submission has been favorably evaluated by Vivek Malhotra (Senior editor) and three reviewers, one of whom, Pekka Lappalainen, is a member of our Board of Reviewing Editors. One of the three reviewers, Liza Pon, has also agreed to share her identity.

The reviewers have discussed the reviews with one another and the Reviewing editor has drafted this decision to help you prepare a revised submission.

Previous studies revealed that constriction of mitochondria, which precedes Drp1-mediated mitochondrial fission, occurs at sites of close contact between ER and mitochondria, and is driven in part by INF2-mediated actin polymerization. Here Manor et al. show that a specific splice-variant of an actin nucleating protein Spire (named Spire1C) localizes to the mitochondria outer membrane through a specific a-helical motif encoded by its alternate exon C. Importantly, they demonstrate that Spire1C promotes actin filament assembly at the mitochondrial surfaces and modulates mitochondrial fission through its actin-binding WH2 domains and its KIND domain, which specifically binds to INF2 formin. Collectively, these studies provide evidence that Spire1C and INF2 cooperate to form mitochondria-ER intersections, and to promote efficient actin filament assembly specifically at these sites.

The majority of the data presented in the manuscript are convincing and the study provides fundamentally important new insights into the mechanisms of mitochondrial fission. However, there are few important issues that should be addressed to confirm the conclusions presented and to further strengthen this study.

Essentials revisions:

1) Most of the studies presented are based on Spire1C over-expression. The authors should thus perform the mitochondrial fission (Figure 4C) and ‘mitochondrial constriction’ assays (Figure 7) also on Spire1C knockdown cells. The effects of Spire RNAi should also be verified with an additional shRNAi construct, or the authors could try to rescue the phenotype of Spire RNAi cells. Together, these experiments would perhaps also reconcile some inconsistencies in the data. For example, while cell quantification suggests that Spire1C overexpression induces mitochondrial fragmentation (Figure 4), the images presented in the manuscript (e.g. in Figure 5) either show normal or even elongated mitochondria.

2) The lipid specificity assay (shown in Figure 2–figure supplement 1) is not particularly convincing and should be omitted from the manuscript. Lipid specificity of a transmembrane protein (or a hydrophobic protein motif that penetrates into the acyl chain region of the bilayer) cannot be studied using lipid strips, where the individual lipid molecules are most likely not organized into a proper bilayer. Thus, if the authors wish to examine lipid specificity of ExonC, they should instead perform proper vesicle co-flotation or co-sedimentation assays.

3) The authors should provide better controls for the protease sensitivity assay. This assay would benefit from analysis of a protease sensitive IMS protein to confirm that the protease treatment conditions used degrades OM protein without affecting the integrity of the OM. Furthermore, based on the sequence analysis (presented in Figure 1–figure supplement 3) ExonC is predicted to consist of two a-helices, which both are long enough (15-20 residues) to span the mitochondrial outer membrane. Would it be possible that these helices could either make a ‘hairpin’, which spans the outer membrane twice, or that this region would instead just ‘horizontally’ penetrate into the acyl-chain region of the mitochondrial outer membrane without spanning the entire bilayer? To provide stronger support for the conclusions presented, the authors should repeat the assay with a C-terminal GFP-fusion of full-length Spire1C (to confirm that the C-terminus of full-length Spire1C indeed does not face the cytosol as proposed in the current version of the manuscript).

---

## [Author Response]

1) Most of the studies presented are based on Spire1C over-expression. The authors should thus perform the mitochondrial fission (Figure 4C) and ‘mitochondrial constriction’ assays (Figure 7) also on Spire1C knockdown cells. The effects of Spire RNAi should also be verified with an additional shRNAi construct, or the authors could try to rescue the phenotype of Spire RNAi cells. Together, these experiments would perhaps also reconcile some inconsistencies in the data. For example, while cell quantification suggests that Spire1C overexpression induces mitochondrial fragmentation (Figure 4), the images presented in the manuscript (e.g. in Figure 5) either show normal or even elongated mitochondria.

We have performed additional experiments using Spire knockdown cells (using two different shRNA constructs), and added the fission and constriction data to the manuscript (Figure 4C and Figure 7A), all of which is consistent with our other data.

With regards to apparent inconsistencies in mitochondrial phenotype in Spire1C overexpressing cells: We found that in all conditions there was a range of mitochondrial lengths within each cell, as well as between cells. While the average mitochondrial lengths significantly changed depending on Spire or INF2 activities, nearly every cell has some mitochondria that are very short, and at least a couple that are longer. For the sake of clarity in some of our experiments (for example the FRAP data in Figure 2, actin accumulation on mitochondria in Figure 3, or the ER-mitochondria intersection data in Figure 5), we used cells with longer mitochondria in order to be able to make measurements as necessary (i.e. photobleaching only a small segment of very short mitochondria is nearly impossible, measuring ER-mitochondria intersections in cells where all the mitochondria are already fragmented is similarly impossible, and actin around fragmented mitochondria is more difficult to detect than on tubulated mitochondria where a continuous line of actin accumulation is more easily detectable within the already large amount of F-actin throughout the cytoplasm). In order to clarify our data in this regard, we have added as a supplemental figure a histogram showing the distribution of mitochondrial lengths for each condition (Figure 4–figure supplement 1).

*2) The lipid specificity assay (shown in Figure 2*–*figure supplement 1) is not particularly convincing and should be omitted from the manuscript. Lipid specificity of a transmembrane protein (or a hydrophobic protein motif that penetrates into the acyl chain region of the bilayer) cannot be studied using lipid strips, where the individual lipid molecules are most likely not organized into a proper bilayer. Thus, if the authors wish to examine lipid specificity of ExonC, they should instead perform proper vesicle co-flotation or co-sedimentation assays.*

We agree that co-flotation and/or co-sedimentation assays would better establish the lipid specificity of ExonC. This will require many more experiments that go beyond the scope of this work. Therefore, we have removed the lipid dot blot and will attempt to establish ExonC’s lipid specificity in a future study.

*3) The authors should provide better controls for the protease sensitivity assay. This assay would benefit from analysis of a protease sensitive IMS protein to confirm that the protease treatment conditions used degrades OM protein without affecting the integrity of the OM. Furthermore, based on the sequence analysis (presented in Figure 1*–*figure supplement 3) ExonC is predicted to consist of two a-helices, which both are long enough (15-20 residues) to span the mitochondrial outer membrane. Would it be possible that these helices could either make a ‘hairpin’, which spans the outer membrane twice, or that this region would instead just ‘horizontally’ penetrate into the acyl-chain region of the mitochondrial outer membrane without spanning the entire bilayer? To provide stronger support for the conclusions presented, the authors should repeat the assay with a C-terminal GFP-fusion of full-length Spire1C (to confirm that the C-terminus of full-length Spire1C indeed does not face the cytosol as proposed in the current version of the manuscript).*

We think the reviewers may have overlooked our control protein in the protease protection assay in making his/her statement “this assay would benefit from analysis of a protease sensitive IMS protein to confirm that the protease treatment conditions used degrades OM protein without affecting the integrity of the OM.” By definition, if our treatment conditions are not affecting the integrity of the OM, then any IMS protein would not be affected. This is exactly why we used the OMI-mCherry construct as a control in all of our experiments. OMI-mCherry localizes to the IMS – if the OM was being degraded by our digitonin treatment, then OMI-mCherry would escape into the cytoplasm. Similarly, if the OM was degraded enough that trypsin was able to digest proteins in the IMS, then OMI-mCherry would be depleted upon addition of trypsin. In none of our experiments did we observe depletion of OMI-mCherry mitochondrial fluorescence, indicating that the OM was indeed intact in all of our experimental conditions.

We agree that it is important to be careful not to make any claims about the C-terminus of Spire1C being in the IMS, since our FPP assay only informs us that the C-terminal region of ExonC (which is in the middle of the full-length Spire protein) is protected from trypsinization. While our FPP assay leaves us confident that the C-terminus of ExonC is protected from the cytoplasm, we were unable to draw any conclusions from these assays as to the localization of the Spire1C C-terminus. While the secondary protein structure prediction software PHYRE identifies two putative alpha-helices within ExonC, only one of those helices was predicted to be a transmembrane domain by predictive software, so we chose not to postulate that there are two transmembrane domains in ExonC. However, we agree that the hairpin model the reviewers have proposed is just as likely to be correct, and is furthermore appealing since this would allow for the C-terminal domain of Spire1C to interact with other cytoplasmic proteins.

We agree that an additional FPP assay using a C-terminus GFP-tagged Spire1C construct would ideally clarify the topology of the Spire1C C-terminus. In order to address the reviewers’ comments, we performed an additional FPP assay using a new Spire1C-GFP construct with GFP on the C-terminus. Unfortunately, when we expressed this construct, the protein no longer localized properly to mitochondria, instead displaying a mostly cytoplasmic localization pattern (with barely detectable accumulation on mitochondria). Due to this disrupted localization pattern, we were unable to draw any conclusions from an FPP assay; upon addition of digitonin, all of the GFP fluorescence was almost immediately depleted, even without subsequent addition of trypsin. While this result may give some clue as to how Spire1C localizes to the OM, many more experiments that go beyond the scope of this work would have to be performed in order to determine what this result is telling us. Thus, we have mentioned these latest findings in the text, and modified our interpretation of our FPP data to include the likely possibility that Spire1C’s ExonC localizes in a hairpin or horizontal fashion on the outer leaflet of the outer mitochondrial membrane.